# The quantum-confined Stark effect in layered hybrid perovskites mediated by orientational polarizability of confined dipoles

G. Walters[1], M. Wei[1], O. Voznyy [1], R. Quintero-Bermudez[1], A. Kiani[1], D.-M. Smilgies [2], R. Munir[3], A. Amassian[3,4], S. Hoogland[1] & E. Sargent [1]

The quantum-confined Stark effect (QCSE) is an established optical modulation mechanism, yet top-performing modulators harnessing it rely on costly fabrication processes. Here, we present large modulation amplitudes for solution-processed layered hybrid perovskites and a modulation mechanism related to the orientational polarizability of dipolar cations confined within these self-assembled quantum wells. We report an anomalous (blue-shifting) QCSE for layers that contain methylammonium cations, in contrast with cesium-containing layers that show normal (red-shifting) behavior. We attribute the blue-shifts to an extraordinary diminution in the exciton binding energy that arises from an augmented separation of the electron and hole wavefunctions caused by the orientational response of the dipolar cations. The absorption coefficient changes, realized by either the red- or blue-shifts, are the strongest among solution-processed materials at room temperature and are comparable to those exhibited in the highest-performing epitaxial compound semiconductor heterostructures.

[1] Department of Electrical and Computer Engineering, University of Toronto, 35 St. George Street, Toronto, ON M5S 1A4, Canada. [2] CHESS Wilson Laboratory, Cornell University, 161 Synchrotron Drive, Ithaca, NY 14853, USA. [3] Physical and Engineering Sciences Division, KAUST Solar Center (KSU), King Abdullah University of Science and Technology (KAUST), Thuwal 23955-6900, Saudi Arabia. [4] Department of Materials Science and Engineering, North Carolina State University, Raleigh, NC 27695, USA. These authors contributed equally: G. Walters, M. Wei. Correspondence and requests for materials should be addressed to E.S. (email: ted.sargent@utoronto.ca)

The ascent of semiconducting hybrid metal-halide perovskite materials as high-efficiency light absorbers has been mirrored by a rise of these materials as bright light emitters. While the perovskites used in photovoltaic devices have been predominantly of the bulk three-dimensional $ABX_3$ form (where A is a cationic group, B is a metal cation, and X is a halide anion), with methylammonium lead-iodide the archetype, the perovskites reported in the highest-performing luminescent devices are typically the low-dimensional forms[1–5]. Large organic cationic ligands that fill the A site can be used to introduce layering within metal-halide perovskites to form bright two-dimensional materials. These cations add energetic barriers that confine charge carriers to quantum wells, and therefore increase radiative recombination rates. The width of the quantum wells, and so the extent of the confinement, can be tuned by judiciously mixing smaller A cations, such as methylammonium or cesium, with the larger organic ligands.

Layered perovskites present a system of quantum wells, self-assembled with many degrees of compositional freedom, that could potentially enable tailoring of the quantum-confined Stark effect (QCSE). The QCSE arises when a quantum-confined system is subjected to an electric field applied along the axis of confinement. For semiconductor quantum wells, the applied field skews the potential well, and this causes the hole and electron energy levels to shift, decreasing the gap between these levels[6–9]. These changes are accompanied by a change to the exciton binding energy due to a reduction of the Coulombic interaction as the electron and hole become spatially separated[6–9]. The well barriers prevent field ionization that would normally occur in unconstrained systems under such large fields. The QCSE usually manifests as a net decrease in energy of the exciton and thus a red-shift of its optical absorption resonance peak[6–9]. The ability to change the optical absorption of confined systems with an electric field has enabled the application of the QCSE in electroabsorption modulators. The best-performing modulators have, until now, relied on single-crystal semiconductors grown via metalorganic vapor phase epitaxy[10–14].

Quantum and dielectric confinement effects in layered perovskites result in exceptionally large exciton binding energies and oscillator strengths that give the exciton resonances marked optical absorption peaks, atypical for conventional semiconducting materials[15–18]. The prominence and sharpness of these absorption features suggest that even small energetic shifts will translate into large changes in absorption. For an exciton resonance with Gaussian broadening, the amplitudes of the absorption changes, $\Delta\alpha$, related to a Stark shift induced by an electric field, $F$, are proportional to the factors that define the shape of the transition's optical absorption, and the alignment of the quantum wells with the electric field. This proportionality is given by:

$$\frac{\Delta\alpha}{F^2} \propto \frac{f \cdot E_B \cdot \cos^2\varphi}{\Gamma} \qquad (1)$$

where the oscillator strength, $f$, and binding energy, $E_B$, determine the amplitude of the optical transition; the linewidth, $\Gamma$, determines its breadth; and the orientational order parameter is defined by the angle, $\varphi$, between the direction of confinement and the applied field. In light of these properties, we hypothesized that layered perovskites could be engineered to produce large modulation amplitudes through the QCSE.

Unlike the fully inorganic semiconductors used in conventional quantum well modulators, hybrid perovskites and their layered derivatives can accommodate dipolar cations. The role of the A cations in bulk perovskites has attracted attention: numerous studies have focused on optimizing mixtures of several cations for improving device performance[19–21], investigating the possibility of alignment of dipolar cations[22–26], and probing the rotational dynamics of dipolar cations[27–32]. Although research into low-dimensional perovskites is growing, the corresponding studies have yet to be conducted in these materials; the differences with the bulk suggest that the smaller cations in low-dimensional perovskites may add intriguing properties. We hypothesized that —since dipolar cations possess an orientational degree of freedom within the quantum well—polarization of the cations could influence the energetic and electronic response of the perovskite to electric fields.

Considering the promising optical and dielectric properties of layered hybrid metal-halide perovskites, we endeavored to investigate their QCSE behavior. Here we report strong QCSE shifts in layered perovskites enabled by cation tuning, and anomalous behavior associated with dipolar cations. When methylammonium cations are incorporated into the perovskite layers, we observe blue-shifts of the exciton resonance peaks, in contrast with cesium-containing perovskites that instead exhibit the conventional red-shifts. We attribute the unusual blue-shifts to large decreases in the exciton binding energy that counteract and dominate over the opposing energy level shifts. The energy decreases result from the reduced electron–hole overlap, amplified by the dipolar polarizability of methylammonium cations. We demonstrate that layered perovskites can be engineered to have QCSE shifts, either to the red or blue, that produce absorption coefficient changes up to $70\ cm^{-1}$ for $56\ kV\ cm^{-1}$ applied electric fields. These represent the largest field-induced changes in absorption coefficient reported for solution-processed materials at room temperature.

## Results

**Layered hybrid perovskite thin films.** The quantum well width of layered perovskites is defined by the number of metal-halide octahedra, $n$, spanning the perovskite region between the barriers of ammonium-terminated organic ligands (Fig. 1a). Most reported layered perovskite systems feature a distribution of well widths in mixed domains, factors that are beneficial for carrier funneling and bright photoluminescence[3–5,33].

In order to investigate the QCSE in layered perovskites, we instead used colloidal perovskite nanoplatelets, which offer finer control over the well width and structural orientation in our devices. We synthesized colloidal lead-bromide nanoplatelets by dropwise addition of perovskite precursor solutions into an antisolvent[34–38]. We were able to produce nearly phase-pure nanoplatelets from $n = 1$ to 3 by varying the relative proportions of the large and small cations used in the precursor solution. Higher $n$ values could be obtained, but always mixed with other phases. Methylammonium, with its large dipole moment of 2.3 D[39], and cesium, with no dipole moment, were used as small cation inclusions. The optical absorption spectra for films of $n = 1$ to 4 nanoplatelets (with hexylammonium ligands and methylammonium cations) show distinct peaks, corresponding to the first exciton resonances characteristic of each well width (Fig. 1b). As the well width decreases, confinement shifts the band-edge to higher energies, and increases the exciton binding energy, oscillator strength, and exciton peak prominence.

QCSE shifts are restricted to systems where electric fields are applied in the direction of confinement. Field components in the plane of the quantum wells promote exciton ionization and lead to broadening of the exciton features.

To study the QCSE in layered perovskites, we required a materials system with well-defined quantum wells, not only in well width, but also structural ordering. Global ordering of the quantum wells and knowledge of their collective orientation is

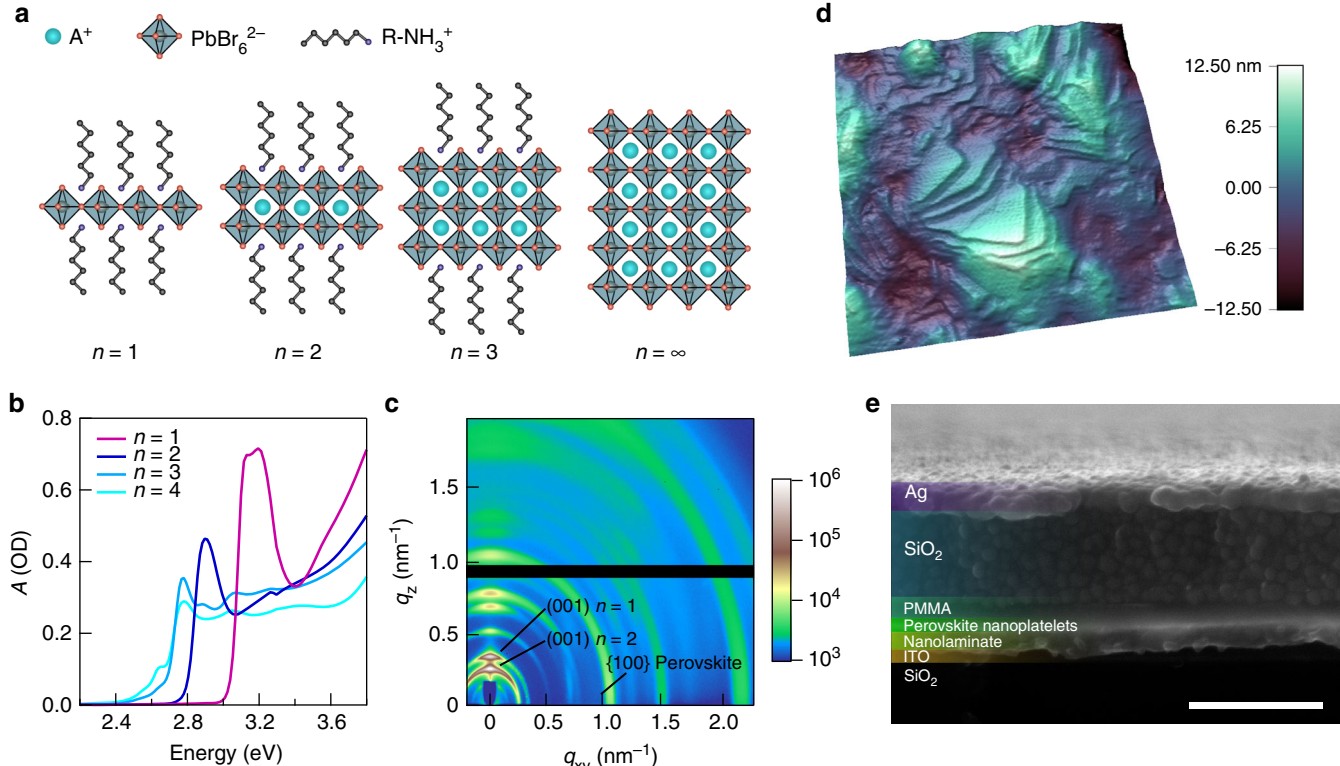

**Fig. 1** Hybrid perovskite nanoplatelets and modulator properties. **a** Illustration of layered hybrid perovskites of varying number, $n$, of lead-halide octahedra spanning the layer. **b** Optical absorption spectra of nanoplatelet thin films for target $n$ values of 1 through 4. **c** GIWAXS plot of intensity and scattering vector of a nanoplatelet film showing preferential alignment of the $c$-axis with the substrate normal. The horizontal black band results from a gap in detector coverage. **d** AFM topograph of a nanoplatelet thin film; lateral length scales are 200 nm. **e** SEM cross-section of a perovskite nanoplatelet electroabsorption modulator. Scale bar indicates 500 nm. The composition of nanoplatelet materials shown follows $(C_6H_{13}NH_3)_2(CH_3NH_3)_{n-1}Pb_nBr_{3n+1}$

necessary to build devices that can establish electric fields solely in the direction of confinement. With this in mind, we used centrifugal casting to deposit phase-pure nanoplatelets as homogeneous and full-coverage thin films. These exhibited a high degree of ordering, exemplified in grazing-incidence wide-angle X-ray scattering (GIWAXS) measurements (Fig. 1c and Supplementary Fig. 1). For a typical $n = 2$ nanoplatelet film, a succession of peaks are found at 90° to the horizontal, corresponding to scattering from the (001) plane (and its harmonics) of the well and barrier layers for an $n = 2$ majority phase, and for an $n = 1$ minority phase. The strength and position of these features, along with a slight ring pattern corresponding to the layers' in-plane {100} scattering, indicate that the nanoplatelets are almost entirely oriented such that their $c$-axis is normal to the substrate. Atomic force microscopy topographs further confirmed the oriented stacking of the nanoplatelets in films. The images revealed that the nanoplatelets have lateral dimensions varying between 10 and 100 nm and stack upon one another with minimal tilting relative to the substrate, forming ~3 nm steps that are consistent with the layer thickness of the target $n = 3$ phase (Fig. 1d and Supplementary Fig. 2).

**Electroabsorption modulation spectroscopy.** Since the nanoplatelets assembled as oriented films, we fabricated modulators in which the perovskite films were sandwiched between electrodes in such a way that the electric field would be established normal to the plane of the platelets, and that QCSEs could therefore be studied (Fig. 1e). We used transparent top (indium tin oxide coated glass) and reflective bottom (Ag) electrodes. Current-blocking layers of nanolaminate formed via atomic layer deposition and spin-cast poly(methylmethacrylate) with

sputtered $SiO_2$ were used to insulate the perovskite from the electrodes. Based on the thicknesses and dielectric properties of the different layers used in these devices, we estimate the electric fields applied to the layered perovskite thin films to be in the range of 50 to 80 kV cm$^{-1}$ (increasing with decreasing $n$ value) for an applied 5 V peak voltage. This corresponds to internal fields in the quantum wells of 10 to 17 kV cm$^{-1}$ (Supplementary Table 1).

Electroabsorption (EA) spectroscopy probes electronic changes in materials in response to applied electric fields. We collected room temperature EA spectra for our nanoplatelet samples by measuring field-induced changes in the intensity of monochromatic light reflected by our modulator devices. The changes in measured reflectance relate to changes in absorption to a first-order approximation as:

$$-\frac{\Delta R}{R} \cong \Delta A = \Delta \alpha d \qquad (2)$$

where $R$ is the reflectance, $A$ is the optical depth, $\alpha$ is the absorption coefficient, and $d$ is the optical path length. The use of a lock-in amplifier referenced to the second harmonic of the modulating bias ensures that the measurements only probe absorption changes associated with the electrical modulation. Spectral features appear as weighted-sums of the zeroth-, first-, and second-derivatives of the optical absorption bands, of which each contribution corresponds respectively to changes in intensity, position, and width of the absorption band. These changes are then interpreted as changes to the transition's oscillator strength, polarizability, and permanent dipole moment, respectively (see Methods for further details)[40]. Although a first-derivative line shape is expected to dominate EA spectra of the

QCSE, second-derivative contributions can appear from either increased tunneling between wells or from exciton ionization in wells misaligned with the applied field.

We first used EA spectroscopy to study the influence of well width on the QCSE in the nanoplatelets. Spectra were acquired for nanoplatelets with phases of $n = 1$ to 4 prepared by controlling the relative amounts of methylammonium cations and hexylammonium ligands used in the colloidal synthesis (Fig. 2). While the $n = 1$ sample shows no EA response and the $n = 2$ shows only a weak response, samples with $n = 3$ and 4 show substantially stronger signatures. A clear correspondence exists between the negative of the first-derivative of the optical absorption spectra and the EA spectra for the $n = 3$ and 4 samples. Even the minority phase of $n = 4$ in the $n = 3$ sample shows this correspondence. The correspondence with the first-derivative spectra is expected and indicates a Stark shift, yet the negative correlation equates to an unusual blue-shifting of the excitonic features.

Intrigued by the blue-shifts, we further investigated the QCSE in the nanoplatelets by engineering further the cationic composition of the perovskites. Substitution of methylammonium with cesium led to samples exhibiting strong red-shifts, including the

$n = 2$ phase (Fig. 3a). The optical changes in nanoplatelets with methylammonium and with cesium, at the respective spectral minimum and maximum, show substantially quadratic dependences on electric field, but with opposing signs (Fig. 3b). Thus strong red- and blue-shifts can be generated in the self-assembled wells by tailoring cation composition. We also explored engineering the length of the ligand cations and therefore the confining barrier. While using a shorter ligand, butylammonium, would not produce stable $n = 3$ films, using the longer ligand octylammonium produced films that showed the same QCSE behavior but with sharper features (Fig. 3c and Supplementary Fig. 9). The closer resemblance to the first-derivative spectrum is due to an improvement in quantum confinement, and so a diminished loss of the excitonic resonance's oscillator strength that results from the separation of the electron and hole; for the hexylammonium samples, this zeroth-derivative contribution produces a slight departure from a pure Stark shift.

**Anomalous QCSE blue-shifts.** Prior EA studies on hybrid perovskites have been limited to bulk methylammonium lead-iodide and pure $n = 1$ layered perovskites. Bulk methylammonium lead-

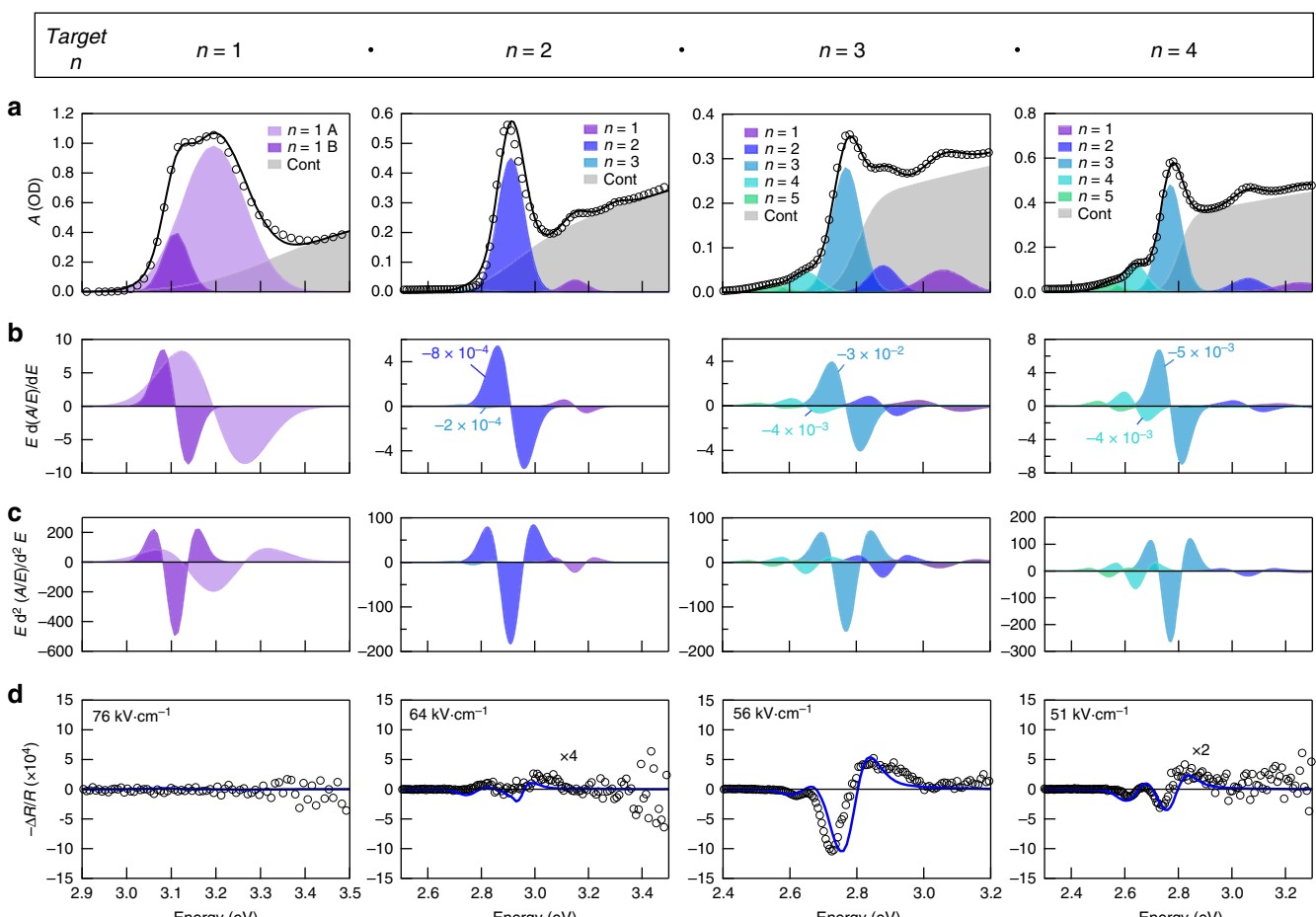

**Fig. 2** Well width-dependent electroabsorption. **a** Optical absorption spectra for perovskite nanoplatelets (methylammonium and hexylammonium organic cations) fit with bound and continuum exciton transitions (see Methods for details). Open circles are experimental data points. The total fit is provided as the solid black line. The target well width increases from left to right, and is labelled above each column. The splitting of the exciton resonance for the $n = 1$ sample is attributed to previously observed phonon sidebands[43,98]. **b** First-derivatives of the excitonic contributions in the optical absorption; correlation with electroabsorption indicates energetic shifts. **c** Second-derivatives of the excitonic contributions in the optical absorption; correlation with electroabsorption indicates broadening. **d** Electroabsorption spectra of each sample (open circles). Spectra for $n = 2$ and $n = 4$ have been enlarged by the factors given. Blue curves are fits from transfer matrix modelling of changes to the complex dielectric function based on weighted-sums of the zeroth-, first-, and second-derivatives of the excitonic absorption bands (see Methods and Supplementary Figures 3–8). The weights of the main contributions are indicated in **b**. Electric field strengths indicate those applied to the nanoplatelet layers

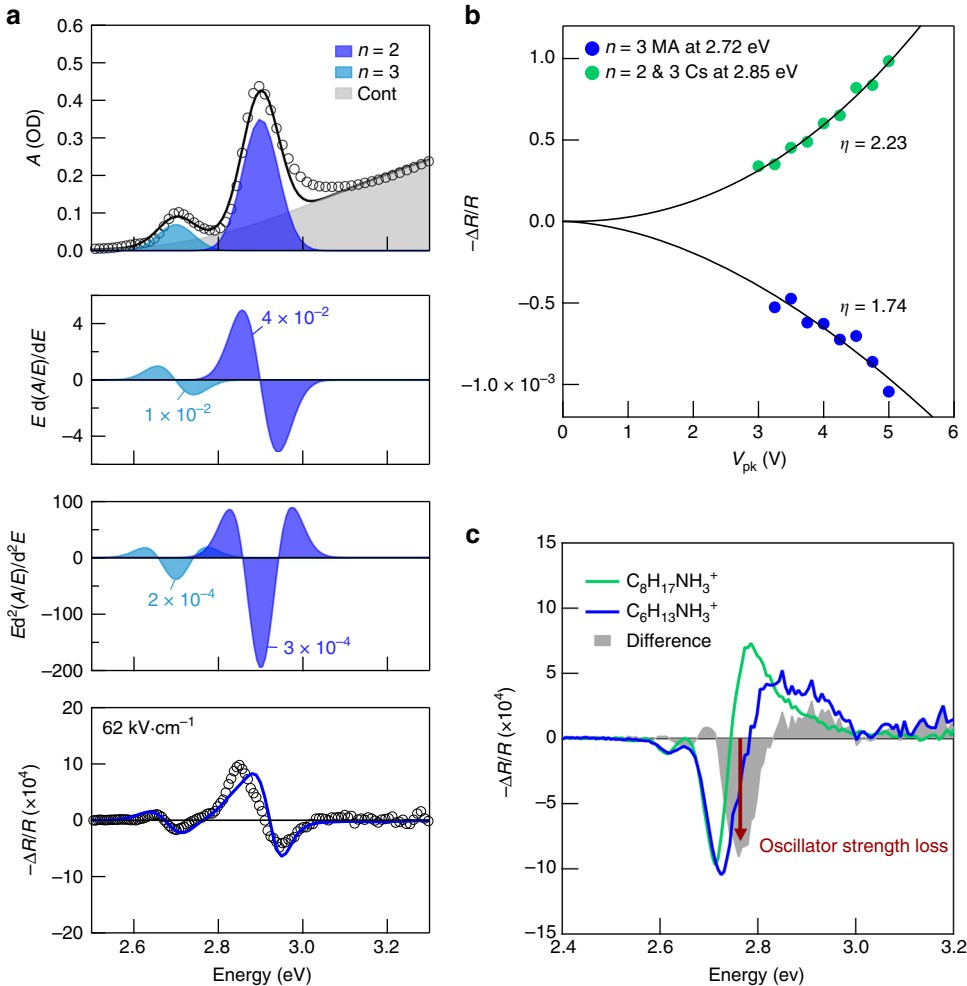

**Fig. 3** Cation tuning of the QCSE. **a** Electroabsorption spectrum for perovskite nanoplatelets with cesium cations. From top to bottom: optical absorption profile and fit, first-derivative of excitonic optical absorption, second-derivative of excitonic optical absorption, electroabsorption spectrum. Open circles indicate experimental data points. Electric field strength indicates that applied to the nanoplatelet layers. **b** Nonlinear dependence of the QCSE on electric field. Solid lines correspond to a power law fit defined by parameter $\eta$. The cesium and methylammonium-based nanoplatelets display opposing reflectance changes at their electroabsorption extrema. **c** Electroabsorption spectra for $n = 3$ methylammonium nanoplatelets with hexylammonium and octylammonium ligands. The difference between the spectra is provided and primarily shows a change in oscillator strength

iodide exhibits a third-derivative EA spectra[41], a result indicative of the Franz–Keldysh–Aspnes (FKA) effect expected for bulk isotropic semiconductors[42] and one we reproduce herein (Supplementary Fig. 10). Several studies on $n = 1$ layered perovskites have revealed similar exciton blue-shifts, but only at low temperatures, where narrowing of the exciton linewidth enables small energetic changes to result in observable absorption changes (5 K)[17,43,44]. These have been attributed to an additional image charge potential generated from the high-frequency dielectric contrast between the lead-halide wells and the organic barriers. The image charges created at the interface promote electron–hole separation and therefore reduce the exciton binding energy. We found, through spectroscopic ellipsometry measurements, that the high-frequency dielectric properties of the cesium and methylammonium-layered perovskites do not differ significantly (Supplementary Figs. 3–8)—as seen in prior studies of their bulk three-dimensional analogues[45–48]. We note that the exciton Bohr radii are similar between the materials with these two cations (Supplementary Table 2). Thus, we hypothesized that a different mechanism must be responsible for the blue-shifts observed in $n > 1$ hybrid perovskites and must be linked to the dipole character of the cations.

For excitons in semiconductor quantum wells, the electron and hole energy levels experience Stark shifts under an electric field that reduce the gap between levels[6–9]. The shifts are accompanied by a typically smaller decrease in exciton binding energy due to decreased Coulombic interaction of the electron and hole as they move to opposing sides of the well[6–9]. The net energetic changes to the exciton resonance are the sum of the energy level shifts and opposing changes to the binding energy. In order for a blue-shift of the exciton resonance to occur, its binding energy must change substantially. To gain further insight into the response of the nanoplatelets to external electric fields, we carried out density functional theory (DFT) and effective mass approximation (EMA) calculations. The EMA calculations show that the Stark shifts to the energy levels of cesium and methylammonium-layered perovskites are similar and on the order of 0.1 meV for $n = 3$ wells under the $10\,\mathrm{kV\,cm^{-1}}$ internal fields used in our experiments (Supplementary Fig. 11). Differences in the energy level shifts due to the cation are not expected given the similarities in the composition of band-edge states between the materials (Supplementary Fig. 12). They are comprised principally of lead and halide states, as widely reported in metal-halide perovskite materials[49,50].

We then considered, using DFT-calculated wavefunctions, the changes in binding energy of the layered perovskites due to electric fields for unpolarized layers, where there was no net alignment of the methylammonium dipoles. The exciton binding energy decreases with increasing applied field but only on the scale of 1 μeV for 10 kV cm$^{-1}$ internal fields (Supplementary Fig. 13), a particularly small amount considering that the binding energies are on the scale of 100's of meV. The changes are also comparable for both cesium and methylammonium containing layers. The experimental energetic changes can be estimated from the EA spectra using $\Delta E = \Delta A \cdot (dA/dE)^{-1}$. The cesium $n = 3$ spectrum reveals that the bandgap must decrease by about 0.3 meV for an applied 5 V peak voltage. The methylammonium $n = 3$ spectrum shows an equivalent but opposing energy shift of about 0.3 meV. Given the strong similarities in band composition between these two materials, we expect that the exciton binding energy must then decrease by at least 0.6 meV in the methylammonium $n = 3$ in order to overcome the 0.3 meV bandgap reduction and produce the observed 0.3 meV blue-shift for 10 kV cm$^{-1}$ internal fields.

Since the methylammonium dipoles possess a degree of orientational flexibility and would be subjected to a torque under the applied electric field, we investigated the impact of the methylammonium dipole mechanics on the exciton binding energy. We describe the overall polarization of the layer due to methylammonium rotation and alignment as $\theta$, the net angular departure of the methylammonium cations from the neutral position (Fig. 4a). We present a color-map of the calculated change in exciton binding energy, using DFT-calculated

wavefunctions, as both a function of applied electric field and methylammonium rotation (Fig. 4b). The plot shows dramatic changes in the binding energy as the methylammonium rotates through its 0 to 90° space. A mere 5 to 10° perturbation to the methylammonium cations is sufficient to account for the experimentally observed blue-shifts. The necessity of only a small perturbation is promising given that a full alignment would be energetically unlikely[28,30,39,51], and that there is a large difference in strength between the dipole field and the externally applied field (Supplementary Fig. 14). As a first approximation, we studied the coupling of the methylammonium dipoles to the applied field through Monte Carlo simulations (Supplementary Fig. 15). These simulations, of a quasi-two-dimensional lattice of interacting dipoles subjected to an electrostatic field, reveal that under the fields expected in our experiments, the net rotation of the methylammonium dipoles is on the order of about one degree. Along with the DFT calculations, they provide a clear order-of-magnitude estimate that only a small perturbation to the net alignment of the methylammonium dipoles is necessary and can be achieved with the fields in our experiments.

As the methylammonium rotates, separation of the electron and hole states is intensified. This can be observed in visualizations of the HOMO and LUMO states. The states of $n = 3$ methylammonium layers, subjected to an internal field of 20 kV cm$^{-1}$, display minimal spatial separation when unpolarized but become widely separated when a net polarization of the methylammonium cations is added (Fig. 4c). Since the band-edge states of these perovskites are restricted to the inorganic framework (Supplementary Fig. 12), the hole and electron states

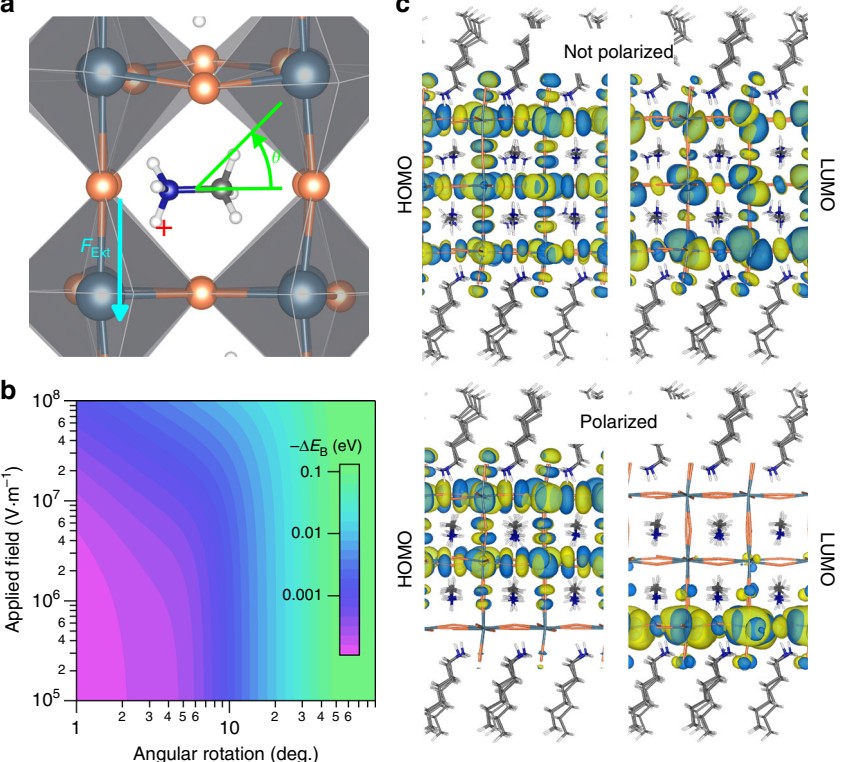

**Fig. 4** Methylammonium influence on changes in the exciton binding energy. **a** Rotation, as defined by $\theta$, of methylammonium cations away from the effective neutral position within a perovskite well when torqued by an external electric field, $F_{Ext}$. A single perovskite unit cell is shown (window size 9.8 × 9.8 Å). **b** Calculated change in exciton binding energy using DFT-calculated wavefunctions as a function of net methylammonium rotation and applied electric field for an $n = 3$ methylammonium-layered perovskite. **c** DFT-calculated spatial reorganization of frontier orbitals in a single $n = 3$ layered perovskite (layer thickness = 34 Å) with unpolarized and polarized methylammonium cations. An electric field of 20 kV cm$^{-1}$ has been applied. The frontier molecular orbitals are shown as isosurfaces (at 0.0046 a.u.) with positive and negative regions of the wavefunctions shown in yellow and blue, respectively

become highly localized on opposing lead-bromide octahedra and thus experience a diminished Coulombic interaction. Based on the reported dynamics for hybrid perovskites[27,29,32,52], we propose that when the electric field is applied to the layered perovskite, the array of polarized dipoles will polarize the exciton.

The observed dependence of the QCSE on the well width of the nanoplatelets requires consideration of several mechanisms. The magnitude of the QCSE shifts of the hole and electron levels has been shown from theory to increase monotonically with well width[53,54]. EMA calculations demonstrate the theoretical quartic dependence of the shifts on well width (Supplementary Fig. 11); however, increases in well width diminish the quantum confinement, such that the exciton may eventually field-ionize and the bulk electric field response would be expected[6,55]. Thus, the QCSE is only relevant on the scale of the exciton Bohr diameter[6]. The reduction in quantum confinement with increasing well width can be shown with calculations of the exciton binding energy based on DFT-calculated wavefunctions (Supplementary Fig. 16), and the difference in binding energy between the bulk and confined systems is a well-documented result[3,56,57]. The polarizability of the hole and electron states is also an important factor. In $n = 1$ materials, the electron and hole states, bound to a single layer of lead-halide octahedra, have no space to be displaced when subjected to an electric field. Alternatively, in the $n > 1$, the states can localize on opposing lead-halide layers. As well, for increasing $n$, the cationic content increases and so has an increasing influence. At around $n = 3$ and $n = 4$, enough methylammonium dipoles are present while the exciton binding energy is still strong, resulting in large changes to the binding energy.

In sum, for cesium wells, changes become measurable at $n = 2$ as the energy level shifts increase with well width; the opposing changes in exciton binding energy are much smaller than the Stark shifts, and so the conventional red-shifting QCSE is observed. In methylammonium wells, the larger opposing changes in exciton binding energy nearly match the Stark shifts in the $n = 2$ sample and overcome those of the $n = 3$ and 4 samples such that a net blue-shifting QCSE is observed.

## Discussion

The dipole-mediated QCSE provides strong optical modulation by exploiting the orientational polarizability and exceptionally strong and narrow optical transitions of hybrid metal-halide perovskites. The strength of the field-induced optical changes observed for our nanoplatelets, which reach absorption coefficient changes of 70 cm$^{-1}$ for 56 kV cm$^{-1}$ applied fields, are the largest reported for thin-film materials at room temperature (Supplementary Table 3). These changes can furthermore be realized as either red- or blue-shifts through simple tailoring of the cation content. Blue-shifting modulators are desirable for their minimized chirp, high optical contrast, and ability to shift transparency towards gain maxima when integrated with a laser[58–60]. However, achieving blue-shifts in conventional compound semiconductors has required complex designs based on asymmetric potentials, superlattice structures, and strain-induced polarizations that only operate for particular field strengths and under unidirectional fields[61–65]. We also note that the optical changes of the QCSE exhibited by perovskite nanoplatelets are roughly one order of magnitude stronger than those for the FKA effect of bulk 3D perovskites—a factor commonly found between the QCSE and FKA effects[66,67]. Despite the self-assembled nature and solution-processed fabrication of the perovskite nanoplatelets, their performance is competitive even with that of inorganic epitaxial semiconductors used in the highest-performing EA modulators (Supplementary Table 3). Future studies could benefit

from in situ investigations of the methylammonium rotation under applied electric fields. This could include the utilization of neutron scattering or Raman scattering spectroscopy techniques to study changes in the rotational dynamics of the methylammonium cations when subjected to external electric fields. Such studies would provide further insights into the dynamic responses of these perovskite materials. Metal-halide perovskites, with their cage-like framework, lend themselves particularly well to dipolar orientational modulation; and the mechanistic principles reported herein can also be deployed in wider materials sets.

## Methods

**Materials and chemical precursors.** Chemicals and materials were purchased from commercial vendors and are as follows: lead(II) bromide, PbBr$_2$, >98% purity, from Alfa Aesar; lead(II) iodide, PbI$_2$, 99.9985% purity, from Alfa Aesar; methylammonium bromide, CH$_3$NH$_3$Br (MABr), from Dyesol; methylammonium iodide, CH$_3$NH$_3$I, from Dyesol; cesium bromide, CsBr, 99.999% purity, from Sigma Aldrich; $n$-hexylammonium bromide, C$_6$H$_{13}$NH$_3$Br (HABr), from Dyesol; $n$-octylammonium bromide, C$_8$H$_{17}$NH$_3$Br (OABr), from Dyesol; n-butylammonium bromide, C$_4$H$_9$NH$_3$Br (BABr), from Dyesol; $N$,$N$-dimethylformamide, C$_3$H$_7$NO (DMF), anhydrous, 99.8% purity, from Sigma Aldrich; chlorobenzene, C$_6$H$_5$Cl (CB), anhydrous, 99.8% purity, from Sigma Aldrich; trimethylaluminum, C$_3$H$_9$Al (TMA), >98% purity, from Strem Chemicals; tetrakis(dimethylamido)zirconium (IV), [(CH$_3$)$_2$N]$_4$Zr (TDMAZ), >99.99% purity, electronic grade, from Sigma Aldrich; tetrakis(dimethylamino)titanium(IV), C$_8$H$_{24}$N$_4$Ti (TDMAT), >99% purity, from Strem Chemicals; poly(methylmethacrylate) (PMMA), average molecular weight 350,000, from Sigma Aldrich; ethyl acetate, C$_4$H$_8$O$_2$ (EtAc), anhydrous, 99.8%, from Sigma Aldrich.

**Perovskite nanoplatelet synthesis.** Colloidal perovskite nanoplatelets were synthesized from solution following a variation of published methods[34–38]. PbBr$_2$, MABr or CsBr, and HABr or BABr or OABr were added to DMF to form a precursor solution. The concentration of PbBr$_2$ was held at 0.06 M, while the ratios of the organic cation components was varied to control the thickness and phase purity of the nanoplatelets. A total of 20 µL of the precursor solution was added dropwise into 1 mL of CB while stirring vigorously. As the perovskite precursors are introduced to the antisolvent, they crystallize immediately as colloidal nanoplatelets capped by the large organic cations present in the precursor solution. A color change is observed from the addition of the precursor solution and bright photoluminescence can be observed when the nanoplatelets are held under UV light. We elected to use lead-bromide for the perovskite inorganic framework because of its greater stability over the other halide analogues.

**Modulator device fabrication.** Modulators consisted of a perovskite layer sandwiched between transparent insulating layers with a transparent electrode on the front-side and a reflecting electrode on the back-side. Indium tin oxide (ITO) coated glass slides were used as the transparent electrode and served as the substrate for the devices. The ITO layer was 30 nm and the glass substrates were 0.8 mm thick. Atomic layer deposition (ALD) was used to deposit alternating layers of Al$_2$O$_3$ and ZrO$_2$ on top of the ITO to prevent current flow between the electrode and the perovskite layer. Each ALD layer involved 20 deposition cycles. Alternating layers were repeated 20 times. Each Al$_2$O$_3$ cycle consisted of a 15 ms pulse of trimethylaluminum with a 7 s purge followed by a 15 ms pulse of H$_2$O with a 7 ms purge. Each ZrO$_2$ cycle consisted of a 100 ms pulse of tetrakis(dimethylamido) zirconium(IV), held at 75°C, with a 7 s purge followed by a 15 ms pulse of H$_2$O with a 7 s purge. A capping layer of 150 cycles of TiO$_2$ was added with ALD. Each TiO$_2$ cycle consisted of a 100 ms pulse of tetrakis(dimethylamino)titanium(IV), held at 75°C, with a 7 s purge followed by a 15 ms pulse of H$_2$O with a 7 s purge. ALD was done with a glovebox integrated Cambridge Nanotech Savannah S100 ALD system with a chamber temperature of 150°C and nitrogen gas flow of 20 sccm. The total thickness of the ALD layers was measured to be 80 nm by AFM and confirmed with SEM. This style of nanolaminate[68] by alternating ALD layers was found to be necessary for effective electrical current blocking and for pinhole prevention. Perovskite nanoplatelets were deposited onto the ALD layers via centrifugal casting. The substrates were placed in a 50 mL centrifuge tube at a tilted angle along with the perovskite solution and centrifuged at 7500 rpm for 15 min. The films were dried in air to remove any remaining solvent residue. The perovskite layers were deposited immediately following synthesis. The perovskite layer was measured to be about 60 nm by AFM. For the 3D bulk MAPbI$_3$ device, the perovskite film was deposited following a reported procedure[33]. An insulating film of poly(methylmethacrylate) (PMMA) was then deposited onto the perovskite film. The PMMA film was prepared by first dissolving PMMA powder in ethyl acetate at a concentration of 3% by weight. The solution was stirred at 800 rpm and 70°C and filtered with a 0.22 µm PTFE filter. The solution was then spin coated at 8000 rpm for 60 s. The PMMA was annealed at 70°C for 10 min. PMMA was deposited immediately after perovskite deposition and drying. The PMMA layer was measured to be about 100 nm by SEM. An ~300 nm insulating layer of SiO$_2$ was then

deposited by sputtering with an Angstrom Engineering sputtering system. A Kurt J. Lesker SiO$_2$ target (99.995% purity) was sputtered at a rate of 0.12 Å s$^{-1}$ with an RF source in nitrogen while the chamber pressure was maintained at 5 mTorr. Samples were rotated with no heating. The combination of SiO$_2$ and PMMA layers was found to be necessary to block electrical current. As well, the PMMA was observed to be necessary to protect the perovskite layer from the sputtering plasma. Overall, 100 nm silver contact pads were deposited by thermal evaporation with an Angstrom Engineering thermal evaporator at a rate of 0.2 Å s$^{-1}$ to serve as the reflecting electrodes. Samples were rotated during evaporation. Each pad's area was 0.07 cm$^2$. Devices were stored in a nitrogen glovebox when not in use. The leakage current through completed devices was measured to be less than 10 nA per pixel. The total device thickness, on top of the glass substrate, was about 670 nm.

**Materials and device characterization.** Optical absorption spectra were measured with a Perkin Elmer 950 UV/VIS/NIR spectrometer equipped with an integrating sphere for thin film measurements. Grazing-incidence wide-angle X-ray spectroscopy (GIWAXS) was conducted at the Cornell High Energy Synchrotron Source (CHESS). The beam was incident to the samples at an angle of 0.5° and had a wavelength of 1.155 Å. The sample-to-detector distance was 173 mm. Atomic force microscopy (AFM) measurements were done with an Asylum Research Cypher operating in tapping mode with an AC240TM-R3 probe. Scanning electron microscopy (SEM) was done with a FEI Quanta FEG 250 ESEM operated under high vacuum at 15 kV. Samples were carbon coated prior to SEM imaging. Spectroscopic ellipsometry was done with a Horiba Jobin Yvon UVISEL Ellipsometer. Films of perovskite nanoplatelets were deposited on glass slides and scotch tape was applied to the rear of the glass to reduce back-reflections. Measurements of the functions $I_s = \sin2\psi\sin\Delta$ and $I_c = \sin2\psi\cos\Delta$ were conducted for three angles of incidence (55°, 65°, 75°) from 1 to 5 eV with a step size of 0.01 eV and integration time of 300 ms.

**Electroabsorption spectroscopy.** Electroabsorption spectra were measured in reflection mode at room temperature in air. To measure $\Delta R$, an AC electric field was applied perpendicular to the perovskite layers with no DC bias by contacting the silver and ITO electrodes with an Agilent 33120a function generator. The sinusoidal AC field was applied with a modulation frequency of 2 kHz. This frequency was selected to ensure measurements were free of hysteretic effects that may be present at low frequencies due to ionic motion[41] and to ensure that the devices were not limited by their RC time constant. White light from a xenon lamp was monochromatised with a grating Triax 320 monochromator, having a slit width of 2 mm and corresponding bandpass of about 5 nm, and focused onto the modulator device. The average irradiance for our spectral region of interest is 380 μW·cm$^{-2}$; the irradiance spectrum is provided in Supplementary Figure 17. Light entered from the glass/ITO side and was reflected by the opposing silver electrode. The reflected light (at 90° to the incident beam) was focused onto a Newport 818-UV/DB photodiode. The AC component from the signal was demodulated using a Stanford Instruments SR830 lock-in amplifier. The lock-in amplifier was phase referenced to the function generator at twice the modulation frequency and was operated in $r$, $\theta$ mode. The second harmonic of the modulation frequency was used to eliminate any electro-optic changes to the spectra (see Supplementary Discussion). The $\Delta R$ spectra were scaled by a factor of $2\sqrt{2}$ to convert from the RMS value. To measure $R$, reflected light was collected from the sample under zero bias. A mechanical chopper was placed in the light path before the sample and was operated at 220 Hz. The lock-in amplifier was referenced to the frequency of the chopper. The $R$ spectra were scaled by a factor of $2\sqrt{2}\pi/4$ to convert from the RMS value. The second-harmonic electroabsorption lock-in technique, we have used is immune to excitation density and built-in field effects as it only detects changes associated with the modulation bias (see Supplementary Discussion). Electroabsorption signals were absent from films of chemical precursors (Supplementary Figure 18). All electroabsorption spectra were found to be repeatable and minimal changes to the response were observed in measurements taken after eight months (Supplementary Figure 19).

**Internal electric field.** The electric field, $F_{layer}$, in a given layer for a device of $n$ adjoining layers with relative permittivities of $\epsilon_n$ and thickness $t_n$ under a potential difference of $V$ is provided by:

$$F_{layer} = \frac{V}{\epsilon_{layer} \sum_1^n \frac{t_n}{\epsilon_n}} \tag{3}$$

The permittivity of the perovskite well is based on the reported values for bulk materials: 25.5 for methylammonium lead-bromide[52], 30 for lead-bromide[69], and 41 for cesium lead-bromide[70]. We set the permittivity of the barrier to be 4.08, based on the permittivity of hexylamine[69]; of PMMA to 3.0[71]; of SiO$_2$ to 4.2[69]; of ZrO$_2$ to 12.5[69]; and of Al$_2$O$_3$ to 9.1[72]. The field strengths over the perovskite and within the wells are provided in Supplementary Table 1.

**Optical absorption fitting.** Optical absorption spectra were fit with a sum of terms accounting for the continuum and bound states of excitons in semiconductor

quantum wells[73–75]. The free electron and hole absorption is influenced by the Coulombic interaction and so is modified with the 2D Sommerfeld enhancement factor. The band-edge absorption can be modelled by the following equation:

$$A(\hbar\omega) = \sum_{n=1}^{\infty} A_n \delta\left(\hbar\omega - E_g + E_B^n\right) + A_0 \cdot \int_0^{\infty} dE \frac{2}{1 - e^{-2\pi\sqrt{\frac{E_g}{E}}}} e^{-\frac{(\hbar\omega-E_g-E)^2}{\Gamma}} \tag{4}$$

where $A_n$ describe the intensity of excitonic transitions, $E_g$ is the bandgap, $n$ is the principal quantum number, $\sigma$ relates the influence of the binding energy on the continuum's band-edge, $\Gamma$ defines the free-electron absorption linewidth, $E_B^n$ equal the exciton binding energies normalized to $n^2$, and $\delta$ is the Dirac-delta function. In order to account for line broadening, the bound exciton states, the first term above, were further convolved with Gaussian functions. Transient absorption spectroscopy measurements aided in the identification of excitonic peaks (see Supplementary Figure 20).

**Electroabsorption line shape.** The electroabsorption curves were interpreted according to the standard formalism of Liptay for absorption bands under an electric field[76,77]. The electroabsorption spectra can be decomposed into contributions corresponding to the zeroth-, first-, and second-derivatives of the absorption bands; each indicating either changes to the oscillator strength, polarizability, or dipole moment. This straightforward treatment serves as an efficient tool for interpreting electroabsorption spectra. Although the formalism is often applied to molecular systems, it is general in describing electroabsorption for systems with discrete absorption bands. Here we provide, briefly, a summary of this treatment concerning the changes in optical absorption corresponding to the energetic shifts of an electronic transition subjected to an electric field.

The energetic shifts, $\Delta\tilde{\nu}_{lk}^F$, to an electronic transition between states $l$ and $k$ subjected to a uniform electric field, **F**, can be treated through Schrodinger perturbation theory such that

$$\Delta\tilde{\nu}_{lk}^F = -\frac{1}{hca}\sum_\alpha \mathbf{F}_\alpha \left[(\boldsymbol{\mu}_\alpha)_{ll} - (\boldsymbol{\mu}_\alpha)_{kk}\right] - \frac{1}{2hca}\sum_\alpha \sum_\beta \mathbf{F}_\alpha \mathbf{F}_\beta \left[(\boldsymbol{\alpha}_{\alpha\beta})_{ll} - (\boldsymbol{\alpha}_{\alpha\beta})_{kk}\right] - \cdots \tag{5}$$

where $\boldsymbol{\mu}_\alpha$ are permanent electric dipole moment components of the states, $\boldsymbol{\alpha}_{\alpha\beta}$ are electric polarizability tensor components of the states, $h$ is Planck's constant, $c$ is the speed of light, $a$ is a conversion constant, and $\alpha$ and $\beta$ are directional indices[77]. The two terms are the first- and second-order Stark shifts.

Light absorption following the Beer–Lambert law possesses a molar absorption coefficient given by

$$\frac{\varepsilon_{lk}(\tilde{\nu})}{\tilde{\nu}} = S\left|\sum_\alpha \mathbf{e}_\alpha(\boldsymbol{\mu}_\alpha)_{lk}\right|^2 s_{lk}^L(\tilde{\nu}) \tag{6}$$

which can be expressed as

$$\frac{\varepsilon(\tilde{\nu})}{\tilde{\nu}} = \frac{1}{3} S \sum_k \sum_l \sum_\alpha w_k (\boldsymbol{\mu}_\alpha)_{lk} (\boldsymbol{\mu}_\alpha)_{kl} s_{lk}(\tilde{\nu}) \tag{7}$$

where $S$ is a constant, $\mathbf{e}_\alpha$ is a component of the unit vector parallel to the electric field of the light, $(\boldsymbol{\mu}_\alpha)_{lk}$ is a component of the transition moment, $w_k$ is the probability of the system being in a given state, and $s_{lk}^L(\tilde{\nu})$ is a line shape function[77]. Under an applied field, this becomes

$$\frac{\varepsilon_\alpha^F(\tilde{\nu})}{\tilde{\nu}} = S \sum_k \sum_l w_k^F \left|(\boldsymbol{\mu}_\alpha^F)_{lk}\right|^2 s_{lk}^F(\tilde{\nu}). \tag{8}$$

From this, we can see that the absorption coefficient can change in three ways: the line shape function, the probabilities of the absorption, and the transition dipole moments. The field dependence of these quantities can then be treated as follows: representing the probability as a canonical ensemble of neighboring energy sublevels and series expanding up to terms quadratic with field, evaluating the transition moment with perturbed wavefunctions, and Taylor series expanding the spectral line shape function. Combining these with the equation for energy shifts, and collecting terms, leads to

$$\varepsilon_\alpha^F(\tilde{\nu}) = \varepsilon(\tilde{\nu})\left[1 + L_\alpha(\tilde{\nu})F^2 + O(F^4)\right] \tag{9}$$

with

$$L_\alpha(\tilde{\nu}) = \frac{\sum_k \sum_l w_k s_{lk}(\tilde{\nu})(L_{\alpha3})_{lk}}{\sum_k \sum_l w_k (\boldsymbol{\pi}_{\alpha\alpha})_{lk} s_{lk}(\tilde{\nu})} \tag{10}$$

$$(L_{\alpha3})_{lk} = (A_{\alpha3})_{lk} + \frac{1}{hca}\left(\frac{d\ln s_{lk}(\tilde{\nu}')}{d\tilde{\nu}'}\right)_{\tilde{\nu}=\tilde{\nu}'}(B_{\alpha3})_{lk} + \frac{1}{2h^2c^2a^2}\left[\left(\frac{d\ln s_{lk}(\tilde{\nu}')}{d\tilde{\nu}'}\right)_{\tilde{\nu}=\tilde{\nu}'}^2 + \left(\frac{d^2\ln s_{lk}(\tilde{\nu}')}{d\tilde{\nu}'^2}\right)_{\tilde{\nu}=\tilde{\nu}'}\right](C_{\alpha3})_{lk} \tag{11}$$

where $\pi_{\alpha\alpha}$ is a transition tensor, and $(A_{\alpha 3})_{lk}$, $(B_{\alpha 3})_{lk}$, and $(C_{\alpha 3})_{lk}$ are spectrally independent coefficients that relate to changes in the transition moment, polarizability, and dipole moment[77]. From this we can see that the spectral dependence of the absorption changes come from a sum of zeroth, first, and second-derivative components. These components can be easily transformed to yield the simple relation from Bublitz and Boxer[40] where the changes in absorption are described by a weighted sum of the zeroth, first, and second derivatives of the optical absorption band,

$$\Delta A(\nu) = F^2 \left[ A_\chi A(\nu) + B_\chi \nu \frac{\mathrm{d}}{\mathrm{d}\nu}\left(\frac{A(\nu)}{\nu}\right) + C_\chi \nu \frac{\mathrm{d}^2}{\mathrm{d}\nu^2}\left(\frac{A(\nu)}{\nu}\right) \right] \quad (12)$$

In doing this the smaller higher order $F^4$ terms have been neglected, all constants have been grouped, and $\Delta A(\nu)$ is the change in optical depth. The coefficient $A_\chi$ corresponds to changes in intensity of the absorption line and reflects field dependence of the transition moment; the coefficient $B_\chi$ corresponds to shifts of the absorption line and reflects field dependence of the polarizability; the coefficient $C_\chi$ corresponds to broadening of the absorption line and reflects field dependence of the dipole moment. Thus, the electroabsorption spectra can be interpreted by analyzing the presence of the different derivative components.

**Optical modelling.** Although the Liptay model can be used to interpret electroabsorption spectra, optical effects inherent to the modulator design can cause deviations in the electroabsorption spectra. We optically model our devices using a transfer matrix model utilizing data obtained from ellipsometry measurements. We find that any optical effects introduced by the devices are minimal (Supplementary Figs. 3–8). This is reinforced by our reproduction (Supplementary Fig. 10) of the line observed for $CH_3NH_3PbI_3$ by Ziffer et al.[41], despite the difference in transmission/reflection measurement set-up.

Spectroscopic ellipsometry data was analyzed using the DeltaPsi2 software from Horiba Jobin Yvon. The optical constants of the perovskite materials were modelled using a Kramers–Kronig consistent series of Voigt oscillators. Models for the samples consisted of a glass substrate with perovskite film of variable thickness and roughness layer of variable thickness. The roughness was modelled with a Bruggeman effective medium layer with 50% void. The $I_s$ and $I_c$ ellipsometry functions were fit using a bound multi-model that bound the material parameters and layer thicknesses across the measurements taken at different angles of incidence. Data were fit over the range of 1.5 to 4.0 eV.

Transfer matrix modelling was done to simulate the operation of the electroabsorption modulators. The modelling procedure followed that outlined by Ziffer et al.[41] Layer thicknesses were set to their experimentally determined values, and the optical constants for the glass, ITO, and perovskites were those found through ellipsometry measurements. The optical constants for $Al_2O_3$, $ZrO_2$, $TiO_2$, PMMA, sputtered $SiO_2$, and Ag were taken from the literature[78–82]. The angle of incidence was set to 45°. Reflectance spectra were calculated for the case of an absent field, and for a field present with $\Delta n$ and $\Delta k$ added to the optical constants for the perovskite layer. $\Delta n$ and $\Delta k$ were calculated from the differential forms of the relations $\varepsilon_r = n^2 - k^2$ and $\varepsilon_i = 2nk$. As Ziffer et al. have provided, the following equations were used:

$$\Delta n = \frac{\Delta\varepsilon_i k + \Delta\varepsilon_r n}{2(n^2 + k^2)} \quad (13)$$

$$\Delta k = \frac{-\Delta\varepsilon_r k + \Delta\varepsilon_i n}{2(n^2 + k^2)} \quad (14)$$

$$\Delta\varepsilon_r(\omega) = \frac{1}{\pi}\int_{-\infty}^{\infty} \frac{\Delta\varepsilon_i(\omega')}{\omega' - \omega}\mathrm{d}\omega' \quad (15)$$

$\Delta\varepsilon_i$ was calculated as a weighted sum of zeroth, first, and second-order derivatives of the Gaussian transitions representing each bound excitonic transition[83]. The simulated electroabsorption spectra were fit to the experimental data by varying the weights of these derivatives.

The possibility of electrostriction was also simulated with optical modelling and it was found that these effects were not present (see Supplementary Figure 21) in our experiments.

**DFT calculations.** Density functional theory (DFT) calculations were done with the CP2K computational package[84] with a mixed Gaussian and plane-wave basis set. Goedecker–Teter–Hutter pseudopotentials[85] in the generalized gradient approximation with the Perdew–Burke–Ernzerhof exchange-correlation functional[86] were used. The MOLOPT basis[87] was used. A grid charge density cut-off of 600 Ry was used. All calculations were done for individual layers of perovskite. An 18 unit supercell with dimensions of $25 \times 25$ Å was used in the plane of the perovskite. This was found to produce a sufficiently randomized arrangement of cations for unpolarized structures, as can be seen by evaluating the exciton binding energy as a function of supercell size (Supplementary Fig. 22). Twenty angstroms of vacuum were added between the

perovskite layers. Molecular dynamics was used to randomize the methylammonium dipoles. Cell dimensions and atomic coordinates were relaxed simultaneously to obtain the unpolarized structures. Methylammnonium cations were manually rotated and then allowed to relax for ten optimization steps to produce the polarized structures. Electric fields were applied along the $c$-axis of the perovskite in strengths ranging from $10^{-5}$ to $10^{-2}$ V Å$^{-1}$. All calculations were done by averaging results from applying the electric field in both the positive and negative directions in order to remove any effects due to extraneous permanent dipoles resulting from the finite size of the simulation volume. Exciton binding energies were calculated by evaluating the Coulomb integrals of the form $\langle \psi_2\psi_1 | (r-r')^{-1} | \psi_1\psi_2 \rangle$ between the HOMO and LUMO orbitals on a 3D grid. Exciton Bohr radii were calculated by evaluating the RMS value of the expectation value for the separation between the HOMO and LUMO orbitals of the form $\sqrt{\langle \psi_2\psi_1 | (r-r')^2 | \psi_1\psi_2 \rangle}$ on a 3D grid[88]. Atomic illustrations were produced by the VESTA software[89].

**EMA calculations.** Calculations of Stark shifts to hole and electron levels were done within the effective mass approximation (EMA) and followed the seminal literature on the QCSE[6,7]. We have reproduced the shifts for the heavy-hole level of the GaAs quantum well reported in these works (see Supplementary Fig. 23). In our calculations we have used an effective mass of 0.3 $m_e$[90]. This has been used for both hole and electron levels and for both cesium and methylammonium perovskites which all show similarity[91–94].

Calculations of the well width dependence were done with an infinite well potential subjected to an electric field. The wavefunctions for such a potential are Airy functions. These calculations are only quantitative for a known effective well width, as the actual well width does not account for penetration of the wavefunctions into the finite barriers of real materials. The energetic shifts in the quantum regime exactly match the theoretical relation given by,

$$\Delta E = \frac{m^* e^2 F^2 L^4}{24\hbar^2 \pi^2}\left(1 - \frac{15}{\pi^2}\right) \quad (16)$$

where $F$ is the field strength, $m^*$ is the effective mass, and $L$ is the well width[54].

Calculations of the Stark shifts regarding the differences due to cations were done with a finite well potential subjected to an electric field and with tailored dielectric responses for the wells, representative of the cesium and methylammonium-based perovskites. These calculations have been done using the tunneling resonance method, where the first resonance in the transmission of a particle across the potential as a function of particle energy corresponds to the lowest energy state. In these calculations, the well and barrier widths were set to 18.6 and 15 Å, the barrier height for the conduction band was set to 0.4 eV and the barrier height for the valence band was set to 2.8 eV[95,96].

**Monte Carlo simulations.** Coupling between the methylammonium dipoles and an applied electric field was studied through numerical simulations using the Monte Carlo method. These simulations were based off of the StarryNight code, available online, that was developed by Aron Walsh's group for describing the rotational dynamics and interactions of methylammonium cations in bulk lead-iodide perovskites[28,97]. The code simulates a three-dimensional cubic lattice of interacting dipoles subjected to an electrostatic field and advances through a Metropolis algorithm. In order to simulate layered perovskites, the periodic boundary conditions in one direction were removed. In addition to the existing energetic interactions of the Hamiltonian (applied electrostatic field interaction, dipole–dipole interactions, and local cage strain term), the interactions of the dipoles with the polarizable dielectric barriers of the quantum wells was included. Dipole–dipole interactions were limited to three lattice units of nearest neighbours. A lattice of $80 \times 80 \times 2$ unit cells was built to simulate the $n = 3$ layered perovskite. The dipoles were initialized with random orientations and were allowed to freely rotate. The simulations were done at a temperature of 300 K and the lattice constant was altered to reflect a lead-bromide lattice. Simulations were done for 4 million Monte Carlo moves, after equilibrating for 100,000 moves. The average angular position of the dipoles was recorded every 20,000 moves, and then averaged to find the net angular rotation for a given field strength.

## Data availability

The data that support the findings of this study are available from the corresponding author upon reasonable request.

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

## Acknowledgements

The work presented in this publication was supported by funding from the Natural Sciences and Engineering Research Council (NSERC) of Canada and from an award (KUS-11-009-21) from the King Abdullah University of Science and Technology. Computations were performed on the General Purpose Cluster supercomputer at the SciNet HPC Consortium. SciNet is funded by: the Canada Foundation for Innovation under the auspices of Compute Canada; the Government of Ontario; Ontario Research Fund—Research Excellence; and the University of Toronto. CHESS is supported by the NSF & NIH/NIGMS via NSF award DMR-1332208. Work was also partially funded by Huawei Canada. We thank E. Palmiano, R. Wolowiec, and D. Kopilovic for assistance in the course of study. We also thank R. Sabatini, L. Quan, O. Ouellette, A. Proppe, J. Fan, M. Saidaminov, A. Jain, M. Liu, and Z. Yang for assistance in the lab and for fruitful discussions.

## Author contributions

G.W., M.W., S.H., and E.H.S. conceived and directed the study. G.W. and M.W. were involved in all aspects of the study and were responsible for any unlisted experimental work and analysis. M.W. designed and synthesized the perovskite materials. G.W., and R.Q.-B. designed and fabricated devices. G.W. did all AFM, EA, SE, and TA measurements. G.W. and O.V. performed the DFT calculations and analysis. G.W. implemented all EMA and MC calculations. A.K. conducted the SEM imaging. D.-M.S., R.M., and A.A. were responsible for the GIWAXS measurements and interpretation. All authors were involved in the preparation of the manuscript.

## Additional information

**Competing interests:** The authors declare no competing interests.

