## [Peer Review file · Nature Communications]

Reviewers' comments:

Reviewer #1 (Remarks to the Author):

In this paper, Walters et al. report the observation of a large quantum confined Stark effect (QCSE) in layered hybrid perovskite "quantum well" structures wherein different QCSE behavior is seen in materials where cesium (Cs) cations vs methylammonium (MA) cations are embedded within the inorganic sheets. In the basic theory of the QCSE, an applied electric field has two simultaneous effects that determine the overall shift in the resonance energy of the exciton. The first is a change in the bandgap due to the reduction of the confinement potential for the single particle electron and hole states, while the second is a change in the exciton binding energy due to the reduction of the electron-hole Coulomb interaction. Here the authors observe a blueshift in the exciton resonance energy with applied field for the MA containing 2D perovskites, while a redshift is observed in the Cs containing materials. The authors attribute the blueshift in the MA perovskites to a decrease in the exciton binding energy with field that is larger in magnitude than the decrease in the bandgap, resulting in an overall blueshift of the exciton resonance for the MA containing perovskite (vs. an overall redshift seen in the Cs containing material where the decrease in the exciton binding energy is smaller than the decrease in the bandgap). The authors attribute the large decrease in binding energy of the MA perovskite to a rotational effect of the MA cation in the applied field. While I do not necessarily agree with the conclusion based on rotation of MA cations to explain the experimental observations and have some further reservations about the analysis, I feel that the manuscript provides a fascinating and possibly important report that the large QCSE in hybrid perovskites can be tuned to blue shift or red shift using simple materials chemistry. Therefore I would recommend this manuscript for consideration in Nature Communications, however not before major revisions to the manuscript have been made (see comments below).

1. My biggest concern is with the conclusion that the blue shift of the exciton resonance in the MA containing perovskite is due to field induced reorientation of the MA cation as opposed to the image charge effect reported by Tanaka et al. (DOI: 10.1103/PhysRevB.71.045312). The study by Tanaka et al is remarkably thorough and quite convincingly shows that the blue shift of the 1s exciton in the electroabsorption spectrum of $(\text{C}_6\text{H}_{13}\text{NH}_3)_2\text{PbI}_4$ perovskite quantum wells is due to an image charge effect related to the dielectric mismatch between the inorganic sheets and the organic barrier ligands in perovskite quantum wells. The authors do indeed mention this study in their manuscript, but quickly dismiss the possibility of image charge effects by claiming that the dielectric properties of CsPbBr_3 and MAPbBr_3 are similar through literature citations. First of all, CsPbBr_3 and MAPbBr_3 are essentially different materials than the ones studied in this manuscript and it is not clear that their dielectric properties (esp. with re. to layer contrast) would be the same as in few layer inorganic perovskite sheets with larger barrier ligands. But more importantly, the authors do not address an essential part of the study by Tanaka et al which shows that excitons of different sizes (1s vs 2s hydrogenic excitons in layered perovskites) undergo different sign QCSE shifts (1s excitons blue shift with field and 2s excitons red shift), and this can be perfectly modeled by the image charge effect. Essentially, an exciton with a smaller radius that is confined more completely in the inorganic sheet (1s) has a comparatively much larger change in binding energy in an applied electric field due to an image potential effect than a larger exciton (2s), whose wavefunction already penetrates further into the surrounding dielectric environment than that of the 1s exciton. This clearly shows the exciton size is crucially important to determining the extent of the image charge effect and whether or not this will lead to a QCSE blue shift or red shift. The authors do not mention any difference in the exciton Bohr radius for Cs containing perovskites vs MA containing perovskites. It seems clear from the Tanaka study that differences in exciton size within the inorganic sheets between the Cs and MA perovskites could completely explain the results in the context of the image charge effect, rather than invoking some new physics about the rotation of MA cations. Can the authors comment at all on the difference in exciton Bohr radius in these two systems? Indeed it is also worth pointing out that a more recent

study by Tony Heinz and coworkers (doi: 10.1103/PhysRevB.92.045414) has also demonstrated that image charge effects (dielectric confinement) has a strong influence on the exciton electronic structure in nanosheets of layered hybrid perovskites. Thus, the authors are in the position of positing new physics by dismissing two fairly thorough studies based on a somewhat dubious assumption.

2. The study by Tanaka et al clearly shows that in modeling the EA spectrum it is important to address the electric field response of all of the excitons responsible for optical absorption. While the authors do decompose their absorption spectra into several exciton features and fit the EA spectrum according to derivatives of the individual excitons, I have some issues with the fitting of the exciton peaks from absorption.

2a) First, the authors write in the SI that the 3D hydrogenic exciton model (Elliott's formula) is used to fit the exciton peaks in the absorption spectrum. This is not the correct model to use for a 2D or pseudo 2D system, rather a 2D hydrogenic model should be used (see the paper by Tanaka and the paper by Heinz and co-workers mentioned in comment 1.).

2b) assigning buried exciton peaks based on fitting a broad room temperature absorption spectrum is questionable. The authors should really consider measuring a low temperature absorption and PL spectrum in order to assign an exciton series.

3. I also have some issues with the model that is used to fit the EA spectrum. In a hydrogenic exciton model (that used by the authors to fit the absorption spectrum), only the s-like exciton states contribute to the optical absorption (see the paper by Heinz or Tanaka). The authors use the Liptay equation typically used to fit the EA spectrum of small molecules to fit the exciton EA in their hydrogenic model, which is not really consistent with the system that they are describing. The Liptay equation fits the EA spectrum to the first and second derivatives of the zero-field absorption spectrum, where the first derivative represents the typical quadratic Stark shift due to the polarizability of the ground and excited states, while the second derivative term is related to field induced shifts due to a permanent dipole moment in the ground and excited states. While the second derivative term is important in molecules, fitting the EA spectrum of a hydrogenic exciton to second derivatives is not appropriate, since s-like states have no permanent dipole moment. Simply put, I wonder if the Liptay equation is appropriate to use given the model the authors use to describe the excitons in their system, and if they may be getting erroneous fits by including second derivatives. In the literature of electro- and photo- modulation spectroscopy on quantum well systems it is typical to use only a first derivative model (see the review paper by Pollak and Shen doi: 10.1016/0927-796X(93)90004-M).

4. My last concern is with the analysis of the EA data taken in reflection geometry.

4a) A few papers have shown that in layered thin film stacks, electromodulation spectra measured in reflection geometry can have significantly distorted spectral features due to thin film interference effects (e.g. doi:10.1103/PhysRevB.92.075201). I am particularly concerned about this based on the device structure used by the authors in this manuscript, where the bottom insulator is based on 20 alternating layers of Al₂O₃ and ZrO₂. I would suggest that the authors use ellipsometry to measure the complex refractive index and complex dielectric function of their materials, decompose these spectra with a hydrogenic exciton model, and then use derivatives of those features in a transfer matrix model to fit their data.

4b) In addition to the effects of purely optical interferences, I also wonder about interference coupled with electrostrictive effects? The perovskites are known to be both very soft materials, and very likely piezoresponsive, and also photostrictive (doi:10.1038/ncomms11193 – that is, they exhibit light-induced lattice changes over 1200 ppm). Again, comparing the data with transfer matrix models as the sample is expanded/contracted might help rule out/confirm such a hypothesis.

Reviewer #2 (Remarks to the Author):

This is a very nice piece of research, carefully described in a paper which was enjoyable to read. The

authors explain clearly - and have demonstrated experimentally - how the quantum-confined Stark effect observed conventionally in inorganic semiconductor multilayers can be transferred to perovskite multilayers formed by solution synthesis. The key point of interest, however, is the anomalous blue-shift of absorption observed in certain perovskite nanostructures, for which the authors advance a hypothesis which they support with theoretical modelling. The hypothesis is very attractive, but the theoretical modelling provides no more than a plausibility argument for this: there are no experimental results which can be directly compared with the DFT calculations reported in Fig 4, so there is no direct evidence that field-induced dipole rotation is responsible for the blue-shift of absorption observed in the methylammonium-based samples. Furthermore, there is no evidence in the paper, neither experimental nor theoretical, that dipole rotation occurs only in the methylammonium perovskites and not in the caesium perovskites. Related to this, I am concerned that fig 4b treats the applied field and the dipole rotation as independent parameters in determining the change in exciton binding energy, whereas the authors argument is that these parameters are strongly coupled. The case for the authors' hypothesis would be strengthened if, at the very least, they calculated the dipole rotation as a function of electric field - and hence the resulting binding energy shift - and did so for both perovskite materials investigated. The authors should also suggest what future experiments could be conducted to confirm their hypothesis.

The figures in the paper require some further clarification:

(i) In fig 1c, what is the significance of the thick black horizontal band? The white text on this figure is extremely difficult to read, even at very high magnification.

(ii) In fig 2, the use of the notation "increasing n" - denoting increasing well width (what values of n?) - combined with use of "n=1,2..." for the different excitonic transitions is rather confusing. The caption says that the experimental absorption data has been "fit" with the excitonic and continuum transitions. How has this fitting been achieved? What are the fitting parameters? Similar comments apply to the electroabsorption "fits" in the 4th row. Please provide the electric field strength for this experiment (and fit) in the figure caption. "Spectra for n=1 and n=4 have been enlarged..." Should this read "n=2 and n=4"? Again this is not clear because the columns have not been labelled. Overall, this is a very neat array of figures, but they are so compressed that it is difficult to see the details.

(iii) In Fig 3, again please state in the figure caption the electric field used to achieve electroabsorption. Please clarify the statement (in the caption) "opposing sign for cesium and methylammonium..."

In Fig 4 (a and c) what are the lengthscales of the regions shown? What is the significance of the colour scale in fig 4c?

Reviewer #3 (Remarks to the Author):

This manuscript details electroabsorption (EA) spectroscopy studies of layered thin films of relatively monodisperse 2D nanoplatelet hybrid perovskites. The results are analyzed in terms of the Quantum-Confined Stark Effect, with the particularly notable result of a blue-shift of the EA spectra that is interpreted as resulting from a field-dependent rotation of the organic cations in the perovskite lattice. This manuscript is very well written, and all of the presented data seems consistent with the highest-quality experimental execution. Further, the results are very interesting, the analysis is well justified, and describes a fundamentally intriguing new insight into this materials system that is already the topic of extensive study in the field. I think that the manuscript is a good fit for this journal and ready for publication. My comments are very minor, and are aimed only to help improve the clarity of the manuscript:

1) In the top left panel of figure 2, I don't think it is ever explained what is the difference between "n = 1A" and "n = 1B".

2) Multiple times in the manuscript, it is indicated that the 2nd harmonic of the perturbing field was

phase locked for analysis. However, the microscopic reason why this was necessary, and what particular other "electro-optic changes to the spectra" (In 383, pg 17) were avoided by not using the first harmonic (or any other harmonic) are unclear. The authors simply state that "the second-harmonic electroabsorption lock-in technique we have used is immune to excitation density and built-in field effects as it only detects changes associated with the modulation bias." Again, it seems that locking-in to any harmonic of the bias modulation would avoid these issues. Perhaps this is standard practice for the EA measurement technique, but the rationalization or justification did not come through in this manuscript.

Reviewer #1 (Remarks to the Author):

In this paper, Walters et al. report the observation of a large quantum confined Stark effect (QCSE) in layered hybrid perovskite “quantum well” structures wherein different QCSE behavior is seen in materials where cesium (Cs) cations vs methylammonium (MA) cations are embedded within the inorganic sheets. In the basic theory of the QCSE, an applied electric field has two simultaneous effects that determine the overall shift in the resonance energy of the exciton. The first is a change in the bandgap due to the reduction of the confinement potential for the single particle electron and hole states, while the second is a change in the exciton binding energy due to the reduction of the electron-hole Coulomb interaction. Here the authors observe a blueshift in the exciton resonance energy with applied field for the MA containing 2D perovskites, while a redshift is observed in the Cs containing materials. The authors attribute the blueshift in the MA perovskites to a decrease in the exciton binding energy with field that is larger in magnitude than the decrease in the bandgap, resulting in an overall blueshift of the exciton resonance for the MA containing perovskite (vs. an overall redshift seen in the Cs containing material where the decrease in the exciton binding energy is smaller than the decrease in the bandgap). The authors attribute the large decrease in binding energy of the MA perovskite to a rotational effect of the MA cation in the applied field. While I do not necessarily agree with the conclusion based on rotation of MA cations to explain the experimental observations and have some further reservations about the analysis, I feel that the manuscript provides a fascinating and possibly important report that the large QCSE in hybrid perovskites can be tuned to blue shift or red shift using simple materials chemistry. Therefore I would recommend this manuscript for consideration in Nature Communications, however not before major revisions to the manuscript have been made (see comments below).

1. My biggest concern is with the conclusion that the blue shift of the exciton resonance in the MA containing perovskite is due to field induced reorientation of the MA cation as opposed to the image charge effect reported by Tanaka et al. (DOI:10.1103/PhysRevB.71.045312). The study by Tanaka et al is remarkably thorough and quite convincingly shows that the blue shift of the 1s exciton in the electroabsorption spectrum of $(\text{C}_6\text{H}_{13}\text{NH}_3)_2\text{PbI}_4$ perovskite quantum wells is due to an image charge effect related to the dielectric mismatch between the inorganic sheets and the organic barrier ligands in perovskite quantum wells. The authors do indeed mention this study in their manuscript, but quickly dismiss the possibility of image charge effects by claiming that the dielectric properties of CsPbBr_3 and MAPbBr_3 are similar through literature citations. First of all, CsPbBr_3 and MAPbBr_3 are essentially different materials than the ones studied in this manuscript and it is not clear that their dielectric properties (esp. with re. to layer contrast) would be the same as in few layer inorganic perovskite sheets with larger barrier ligands. But more importantly, the authors do not address an essential part of the study by Tanaka et al which shows that excitons of different sizes (1s vs 2s hydrogenic excitons in layered perovskites) undergo different sign QCSE shifts (1s excitons blue shift with field and 2s excitons red shift), and this can be perfectly modeled by the image charge effect. Essentially, an exciton with a smaller radius that is confined more completely in the inorganic sheet (1s) has a comparatively much larger change in binding energy in an applied electric field due to an image potential effect than a larger exciton (2s), whose wavefunction already penetrates further into the surrounding dielectric environment than that of the 1s exciton. This clearly shows the exciton size is crucially important to determining the extent of the image charge effect and whether or not this will lead to a QCSE blue shift or red shift. The authors do not mention any difference in the exciton Bohr radius for Cs containing perovskites vs

MA containing perovskites. It seems clear from the Tanaka study that differences in exciton size within the inorganic sheets between the Cs and MA perovskites could completely explain the results in the context of the image charge effect, rather than invoking some new physics about the rotation of MA cations. Can the authors comment at all on the difference in exciton Bohr radius in these two systems? Indeed it is also worth pointing out that a more recent study by Tony Heinz and coworkers (doi: 10.1103/PhysRevB.92.045414) has also demonstrated that image charge effects (dielectric confinement) has a strong influence on the exciton electronic structure in nanosheets of layered hybrid perovskites. Thus, the authors are in the position of positing new physics by dismissing two fairly thorough studies based on a somewhat dubious assumption.

We now provide calculations of the exciton Bohr radii for all materials presented in this study. These calculations are based on DFT-calculated wavefunctions. The results are given in the table below.

Exciton binding energy is calculated using the Coulomb scattering integral of the form $\langle \psi_2 \psi_1 | (r - r')^{-1} | \psi_1 \psi_2 \rangle$.

The RMS Bohr radius is calculated using the integral of the form $\sqrt{\langle \psi_2 \psi_1 | (r - r')^2 | \psi_1 \psi_2 \rangle}$ [Arora *et al. Nat. Comms.* 639, 2017].

Material (n value, MA/Cs, ligand)	Bohr Radius (Å)	Exciton Binding Energy (meV)
n = 1, C6	14.09	644
n = 2, MA, C6	15.14	540
n = 2, Cs, C6	14.96	547
n = 3, MA, C6	16.41	476
n = 3, Cs, C6	15.68	507
n = 4, MA, C6	16.99	460
n = 3, MA, C8	16.39	478

The Bohr radius for the n = 1 perovskite is in close agreement with that provided in the literature for similar two-dimensional perovskites [Tanaka *et al. Jpn. J. Appl. Phys.* 44 (8), p. 5923, 2005; Tanaka *et al. Phys. Rev. B.* 71, 045312, 2005; Muljarov *et al. Phys Rev. B.* 51 (20), 14370, 1995]. The difference in radius between the layered perovskites with Cs and MA cations is less than 5%. This is further evidenced in the observation that the positions of the exciton resonances for these materials are nearly identical, and so they must have similar confinement and exciton localization. In contrast, the 1s and 2s excitons of layered perovskites, which do show different QCSE behaviors, have been observed to have quantum confinement enhancement factors >6 [Tanaka *et al. Solid State Communications*, 122, p.249, 2002]. We conclude that the change in sign of the QCSE shifts is not due to differences in exciton Bohr radii between the cesium and methylammonium compounds.

Considering the ellipsometry measurements we carried out in response to the Reviewer's point #4 (Supplementary Figures 3 to 8), we see that the high frequency dielectric constants do not vary significantly between the different layered perovskites. Since the barrier material is the same amongst the cesium and methylammonium samples, the dielectric properties of the wells

themselves must be similar. Thus, we conclude that the differences in the image charge potential created by the contrast of the high frequency dielectric properties between the cesium and methylammonium samples are minimal and cannot explain the change in sign of the QCSE shifts.

We furthermore note that the observed and theoretical blue shifts reported [Tanaka *et al. Phys. Rev. B.* 71, 045312, 2005; Tanaka *et al. Jpn. J. Appl. Phys.* 44 (8), p. 5923, 2005] for $n = 1$ perovskites produced by the image charge potential display a field dependence that does not agree with the trend we observe in Figure 3b. These prior reports show a field-dependence for which a maximum blue-shift can be reached such that eventually red-shifts would occur at high fields. Our data indicate a monotonic and accelerating dependence on field strength, dissimilar to that for image charge effects.

We now note these points in the main text and include the above table as Supplementary Table 2.

2. The study by Tanaka et al clearly shows that in modeling the EA spectrum it is important to address the electric field response of all of the excitons responsible for optical absorption. While the authors do decompose their absorption spectra into several exciton features and fit the EA spectrum according to derivatives of the individual excitons, I have some issues with the fitting of the exciton peaks from absorption.

2a) First, the authors write in the SI that the 3D hydrogenic exciton model (Elliott's formula) is used to fit the exciton peaks in the absorption spectrum. This is not the correct model to use for a 2D or pseudo 2D system, rather a 2D hydrogenic model should be used (see the paper by Tanaka and the paper by Heinz and co-workers mentioned in comment 1.).

We now model the absorption spectra using a 2D model accounting for the bound and continuum states of excitons in semiconductor quantum wells [Winkler, *Phys. Rev. B.* 51 (20), p.14395, 1995; Chuang et al., *Phys. Rev. B.* 43 (2), p.1500, 1991]. We model the bandedge absorption by the following equation:

$$A(\hbar\omega) = \sum_{n=1}^{\infty} A_n \delta(\hbar\omega - E_g + E_B^n) + A_0 \cdot \int_0^{\infty} dE \frac{2}{1 - e^{-2\pi\sqrt{\frac{\sigma}{E}}}} e^{-\frac{(\hbar\omega - E_g - E)^2}{\Gamma}}$$

where A_n describe the intensity of excitonic transitions, E_g is the bandgap, n is the principal quantum number, σ relates the influence of the binding energy on the continuum bandedge, Γ defines the free-electron absorption linewidth, E_B^n equal the exciton binding energies normalized to n^2 , and δ is the Dirac-delta function. In order to account for line broadening, the bound exciton states, the first term above, were further convolved with Gaussian functions.

The conclusions of the work remain unaffected by this change.

The main text and supplementary figures of optical absorption have been updated with fits produced from this model and the supplementary methods have been updated.

2b) assigning buried exciton peaks based on fitting a broad room temperature absorption spectrum is questionable. The authors should really consider measuring a low temperature absorption and PL spectrum in order to assign an exciton series.

Layered perovskites with alkylammonium ligands are known to undergo a phase transition below room temperature related to the ordering of the alkylammonium chains. The transition alters the crystal structure such that the exciton resonances significantly shift [Yaffe *et al. Phys. Rev. B.* 92, 045414, 2015]. Rather than measure low-temperature spectra, we have instead measured transient absorption traces, which show clearly defined ground state bleaches corresponding to the exciton resonances for the different quantum well widths. We have captured the spectra with a 3 picosecond delay between the pump and probe traces for target n values and provide them in the plot below.

We include this plot as Supplementary Figure 22 and have added measurement details to the Supplementary Methods.

3. I also have some issues with the model that is used to fit the EA spectrum. In a hydrogenic exciton model (that used by the authors to fit the absorption spectrum), only the s-like exciton states contribute to the optical absorption (see the paper by Heinz or Tanaka). The authors use the Liptay equation typically used to fit the EA spectrum of small molecules to fit the exciton EA in their hydrogenic model, which is not really consistent with the system that they are describing. The Liptay equation fits the EA spectrum to the first and second derivatives of the zero-field absorption spectrum, where the first derivative represents the typical quadratic Stark shift due to the polarizability of the ground and excited states, while the second derivative term is related to field induced shifts due to a permanent dipole moment in the ground and excited states. While the second derivative term is important in molecules, fitting the EA spectrum of a hydrogenic exciton to second derivatives is not appropriate, since s-like states have no permanent dipole moment. Simply put, I wonder if the Liptay equation is appropriate to use given the model the authors use to describe the excitons in their system, and if they may be getting erroneous fits by including second derivatives. In the literature of electro- and photo-modulation spectroscopy on quantum well systems it is typical to use only a first derivative model (see the review paper by Pollak and Shen doi:10.1016/0927-796X(93)90004-M).

We now better make the case in the revised work that - though the Liptay model is often applied for molecular systems - the theory is general in describing electroabsorption for systems with discrete absorption bands. We thereby provide support that it is a useful tool in interpreting the EA spectra of confined semiconductor systems, and that this has been done in related prior studies [Colvin *et al. J. Chem. Phys.* 101 (8), p.7122, 1994; Liu *et al. App. Phys. Lett.* 98, p.160911, 2011].

Nevertheless, in revising our fitting procedure in accordance with the Reviewer's point #4, we now base our fits from changes to the imaginary dielectric function according to a paper by Pollak and others [Huang *et al. J. Appl. Phys.* 70 (7), p.3808, 1991]. In this work – referenced in Pollak's review paper for providing the line shapes for excitonic transitions with Gaussian broadening – the second derivative term is included to describe broadening to the transition.

While the energetic shifts produced by the QCSE and the consequent first-derivative line shape are expected to dominate the EA spectra for the QCSE, second-derivative line shapes can appear either from increased tunneling between wells when the field is applied, or, if wells are misaligned with the applied field, exciton ionization due to components of the electric field being applied in the plane of the wells [Huang *et al. J. Appl. Phys.* 70 (7), p.3808, 1991]. We now make a note of this in the main text. Additionally, solution-processed quantum confined systems have shown dominating second derivative line shapes that result from dipoles created by imperfections [Colvin *et al. J. Chem. Phys.* 101 (8), p.7122, 1994; Liu *et al. App. Phys. Lett.* 98, p.160911, 2011]. Although our EA spectra plainly exemplify first derivative line shapes, we have included the second derivative for the sake of completeness and, particularly in the Figure panels, to illustrate that our materials are highly ordered and of high quality.

4. My last concern is with the analysis of the EA data taken in reflection geometry.

4a) A few papers have shown that in layered thin film stacks, electromodulation spectra measured in reflection geometry can have significantly distorted spectral features due to thin film interference effects (e.g. doi:10.1103/PhysRevB.92.075201). I am particularly concerned about this based on the device structure used by the authors in this manuscript, where the bottom insulator is based on 20 alternating layers of Al₂O₃ and ZrO₂. I would suggest that the authors use ellipsometry to measure the complex refractive index and complex dielectric function of their materials, decompose these spectra with a hydrogenic exciton model, and then use derivatives of those features in a transfer matrix model to fit their data.

We now provide full optical modelling using experimentally-measured optical constants and the transfer matrix method to simulate electroabsorption and optical effects in our modulators. We followed the approach of Ziffer *et al. [ACS Photonics*, 3, p. 1060, 2016] We provide plots of the ellipsometry data, complex refractive index, complex dielectric function, simulated reflectance spectrum, changes to the complex refractive index, and simulated electroabsorption curve in Supplementary Figures 3 to 8 for each of the perovskite materials we have presented. We also detail the ellipsometry and transfer matrix methods in the Supplementary Methods. We have also updated the electroabsorption fits in the main text figures to reflect this modelling.

We find that while any interference effects in our devices or the changes to the perovskite refractive indices accompanying the electroabsorption do produce subtle changes to the shapes of the

expected electroabsorption curves, the changes are minor and do not affect any of the conclusions drawn from the electroabsorption data. We also note that, despite measuring in reflection mode, we reproduce the electroabsorption line shapes (Supplementary Figure 10) observed by Ziffer *et al.* for $\text{CH}_3\text{NH}_3\text{PbI}_3$ who obtained their measurements in transmission mode. This evidence supports the assertion that the electroabsorption spectra in our manuscript are not significantly influenced by optical effects from the device layers.

4b) In addition to the effects of purely optical interferences, I also wonder about interference coupled with electrostrictive effects? The perovskites are known to be both very soft materials, and very likely piezoresponsive, and also photostrictive (doi:10.1038/ncomms11193 – that is, they exhibit light-induced lattice changes over 1200 ppm). Again, comparing the data with transfer matrix models as the sample is expanded/contracted might help rule out/confirm such a hypothesis.

We now provide a sensitivity analysis of the simulated electroabsorption curves coupled with electrostrictive effects for an $n = 3$ methylammonium perovskite device. We have carried out simulations in which the film thickness is altered for the reflectance spectra in which a field is applied. The plot on the left shows the purely electrostrictive case with no electroabsorption changes included in the simulation. The right plot shows the coupled case, where we have simulated both electrostriction and electroabsorption. We have plotted for lattice expansions or contractions of up to 1200 ppm. The nominal trace indicates the purely electroabsorptive case, as is reported in the manuscript.

We observe that the lineshape produced by electrostriction does not resemble the experimental lineshape we have measured. In fact, electrostriction is weakest where electroabsorption would occur. Even considering the coupled case, we do not observe the electrostriction signatures that would appear at 2.5 eV or at high energies. Thus, we conclude that electrostrictive effects in our films are minimal and do not affect the electroabsorption spectra.

We provide these plots now as Supplementary Figure 21.

Reviewer #2 (Remarks to the Author):

This is a very nice piece of research, carefully described in a paper which was enjoyable to read. The authors explain clearly - and have demonstrated experimentally - how the quantum-confined

Stark effect observed conventionally in inorganic semiconductor multilayers can be transferred to perovskite multilayers formed by solution synthesis. The key point of interest, however, is the anomalous blue-shift of absorption observed in certain perovskite nanostructures, for which the authors advance a hypothesis which they support with theoretical modelling. The hypothesis is very attractive, but the theoretical modelling provides no more than a plausibility argument for this: there are no experimental results which can be directly compared with the DFT calculations reported in Fig 4, so there is no direct evidence that field-induced dipole rotation is responsible for the blue-shift of absorption observed in the methylammonium-based samples. Furthermore, there is no evidence in the paper, neither experimental nor theoretical, that dipole rotation occurs only in the methylammonium perovskites and not in the caesium perovskites. Related to this, I am concerned that fig 4b treats the applied field and the dipole rotation as independent parameters in determining the change in exciton binding energy, whereas the authors' argument is that these parameters are strongly coupled. The case for the authors' hypothesis would be strengthened if, at the very least, they calculated the dipole rotation as a function of electric field - and hence the resulting binding energy shift - and did so for both perovskite materials investigated. The authors should also suggest what future experiments could be conducted to confirm their hypothesis.

We now provide Monte Carlo simulations that show the coupling between the applied electric field and the induced methylammonium rotation. These calculations are based on those developed by Aron Walsh for bulk methylammonium perovskites [Leguy et al. *Nat. Comms.* 6:7124, 2015; Frost et al. *APL Materials*, 2, 081506, 2014]. A three dimensional lattice of interacting dipoles is subjected to an electric field. As the Monte Carlo simulation advances, reorientations of the dipoles are evaluated using a Metropolis algorithm. We have modified the authors' code, available online, to simulate our layered perovskites.

We now include a supplementary figure (S15) showing the net rotation of the methylammonium cations in an $n = 3$ perovskite as a function of applied field strength and reference this in the main text.

We see that for the fields applied to the layered perovskites in our experiments ($56 \text{ kV} \cdot \text{cm}^{-1}$), the net alignment of the methylammonium dipoles is on the order of one degree. This supports DFT

findings indicating that only a slight perturbation to the methylammonium orientations is needed to produce the necessary energetic changes for a blue-shift.

Details of the Monte Carlo simulations are now provided in the Supplementary Materials.

Since cesium is isotropic and possesses no dipole moment, there is no rotation available under an applied field. Although the cesium perovskite could still display local polar fluctuations of its lattice, these have been shown, through low-frequency Raman scattering, to be similar in both the methylammonium and cesium perovskites and to occur regardless of the cation choice [Yaffe *et al. Phys. Rev. Lett.*, 118, p. 136001, 2017].

Future experiments could focus on *in-situ* investigations of the methylammonium rotation under applied electric fields. This could include the utilization of neutron scattering or Raman scattering spectroscopy to study changes in the rotational dynamics of the methylammonium cations when subjected to external electric fields. The application of strong electric fields within either experimental set-up is a great technical challenge, yet surmounting this difficulty would surely allow for interesting insights into the dynamic responses of these perovskite materials. We now note these opportunities in the main text.

The figures in the paper require some further clarification:

(i) In fig 1c, what is the significance of the thick black horizontal band? The white text on this figure is extremely difficult to read, even at very high magnification.

The horizontal black band results from a gap in detector coverage offered by the instrument. We now note this in the figure caption.

The text labels have been replaced with tags that have larger black and bold-faced text.

(ii) In fig 2, the use of the notation "increasing n" - denoting increasing well width (what values of n?) - combined with use of "n=1,2..." for the different excitonic transitions is rather confusing. The caption says that the experimental absorption data has been "fit" with the excitonic and continuum transitions. How has this fitting been achieved? What are the fitting parameters? Similar comments apply to the electroabsorption "fits" in the 4th row. Please provide the electric field strength for this experiment (and fit) in the figure caption. "Spectra for n=1 and n=4 have been enlarged..." Should this read "n=2 and n=4"? Again this is not clear because the columns have not been labelled. Overall, this is a very neat array of figures, but they are so compressed that it is difficult to see the details.

We have now removed the header "increasing n" in Figure 2 and replaced it with labels of the target n value for each sample's column.

Revisions to the fitting procedure for both the absorption data and the electroabsorption data have been implemented in accordance with the recommendations of Reviewer 1.

The absorption data are now fit using a 2D hydrogenic model for excitonic semiconductors [Winkler, *Phys. Rev. B.* 51 (20), p.14395, 1995; Chuang et al., *Phys. Rev. B.* 43 (2), p.1500, 1991]:

$$A(\hbar\omega) = \sum_{n=1}^{\infty} A_n \delta(\hbar\omega - E_g + E_B^n) + A_0 \cdot \int_0^{\infty} dE \frac{2}{1 - e^{-2\pi\sqrt{\frac{\sigma}{E}}}} e^{-\frac{(\hbar\omega - E_g - E)^2}{\Gamma}}$$

where A_n describe the intensity of excitonic transitions, E_g is the bandgap, n is the principle quantum number, σ relates the influence of the binding energy on the continuum's bandedge, E_B^n equal the exciton binding energies normalized to n^2 , and δ is the Dirac-delta function. The two terms in the above equation represent the continuum and bound states for quantum well excitons. In order to account for line broadening, the bound exciton states, the first term above, were convolved further with Gaussian functions.

We now optically model our devices and use this model to obtain fits to the electroabsorption data, which relies on a weighted sum of the zeroth, first, and second derivatives of the absorption profiles for the excitonic transitions.

Further details of fitting for the absorption and electroabsorption data are given in the revised Supplementary Methods.

The electric field strength for each experiment is now indicated in each of the electroabsorption plots.

The caption should read “n=2” and “n=4”. We have corrected the caption to reflect this.

(iii) In Fig 3, again please state in the figure caption the electric field used to achieve electroabsorption. Please clarify the statement (in the caption) "opposing sign for cesium and methylammonium..." In Fig 4 (a and c) what are the lengthscales of the regions shown? What is the significance of the colour scale in fig 4c?

The electric field strength for the experiment is now indicated in the electroabsorption plot.

The caption statement has been amended to “The cesium and methylammonium based nanoplatelets display opposing reflectance changes at their electro-absorption extrema.”

A window size of $9.8 \times 9.8 \text{ \AA}$ shows the single perovskite unit cell. This is now indicated in the figure caption.

The thickness of the layered perovskite in Fig 4c is now provided in the caption (34 \AA).

The following statement has been added to the caption regarding the colour scale of the frontier orbitals: “The frontier molecular orbitals are shown as isosurfaces (at 0.0046 a.u.) with positive and negative regions of the wavefunctions shown in yellow and cyan, respectively”.

Reviewer #3 (Remarks to the Author):

This manuscript details electroabsorption (EA) spectroscopy studies of layered thin films of relatively monodisperse 2D nanoplatelet hybrid perovskites. The results are analyzed in terms of the Quantum-Confined Stark Effect, with the particularly notable result of a blue-shift of the EA spectra that is interpreted as resulting from a field-dependent rotation of the organic cations in the perovskite lattice. This manuscript is very well written, and all of the presented data seems consistent with the highest-quality experimental execution. Further, the results are very interesting, the analysis is well justified, and describes a fundamentally intriguing new insight into this materials system that is already the topic of extensive study in the field. I think that the manuscript is a good fit for this journal and ready for publication. My comments are very minor, and are aimed only to help improve the clarity of the manuscript:

1) In the top left panel of figure 2, I don't think it is ever explained what is the difference between "n = 1A" and "n = 1B".

We now include references in the figure caption to prior observations that the separation of the peaks near the exciton resonance in the n = 1 perovskite result from phonon sidebands [Straus *et al. J. Am. Chem. Soc.*, 138, p.13798, 2016; Tanaka *et al. Phys. Rev. B.* 71, 045312, 2005].

2) Multiple times in the manuscript, it is indicated that the 2nd harmonic of the perturbing field was phase locked for analysis. However, the microscopic reason why this was necessary, and what particular other "electro-optic changes to the spectra" (ln 383, pg 17) were avoided by not using the first harmonic (or any other harmonic) are unclear. The authors simply state that "the second-harmonic electroabsorption lock-in technique we have used is immune to excitation density and built-in field effects as it only detects changes associated with the modulation bias." Again, it seems that locking-in to any harmonic of the bias modulation would avoid these issues. Perhaps this is standard practice for the EA measurement technique, but the rationalization or justification did not come through in this manuscript.

We now better explain, and reference, the use of a lock-in amplifier referenced to the second harmonic of the modulation frequency. This is used in electroabsorption measurements to avoid spurious effects from linear dependences on the modulation field.

Within the low-field regime, electroabsorptive effects vary quadratically with field strength, whereas linear electro-optic or electromechanical effects such as the Pockels effect or piezoelectric effect vary linearly with field strength [Aspnes, *Surface Science.* 37, 418, 1973; Kyser and Rehn, *Solid State Communications*, 8, p.1437, 1970]. These effects may produce changes to the reflectance spectra of electroabsorption modulators and therefore appear as absorption changes. If we group these undesired effects under A_1 and the electroabsorptive effects under A_2 , then,

$$\Delta A = A_1(\nu)F + A_2(\nu)F^2 .$$

The electric field used in modulation spectroscopy can be written as,

$$F_{Tot} = F_{App} \sin(\omega t)$$

where ω is the modulation frequency. The square of this is then,

$$F^2 = F_{App}^2 \sin^2(\omega t)$$

which is equivalent to,

$$F^2 = F_{App}^2 \left[\frac{1 - \cos(2\omega t)}{2} \right].$$

The total absorption/reflection changes, ΔA , can then be written as:

$$\Delta A = A_1(\nu) F_{App} \sin(\omega t) + A_2(\nu) F_{App}^2 \left[\frac{1 - \cos(2\omega t)}{2} \right]$$

By locking into the second harmonic of the modulation frequency, only changes related to the electroabsorptive effects will be detected.

Similarly, if the device had a built in field, F_{Dev} , the total field would be,

$$F_{Tot} = F_{App} \sin(\omega t) + F_{Dev}$$

Again considering the quadratic dependence of electroabsorptive changes on field,

$$F^2 = F_{App}^2 \left[\frac{1 - \cos(2\omega t)}{2} \right] + F_{Dev}^2 + 2F_{App} F_{Dev} \sin(\omega t)$$

we see that only electroabsorptive changes related to the modulation bias will be detected when locking into the second harmonic.

We have now added these discussions to the Supplementary Methods.

REVIEWERS' COMMENTS:

Reviewer #1 (Remarks to the Author):

In this revised manuscript, the authors addressed the major issues in my last review with additional experiments, calculations and analyses listed below. Upon these revisions, I find their conclusion that the blue shift of the exciton resonance in the methylammonium (MA) containing perovskite is due to field induced reorientation of the MA cation well-supported. Therefore, I recommend this manuscript for publication in Nature Communications.

1. The authors addressed my concerns on attributing the exciton resonance blue-shift solely to the MA polarization without examining the MA and Cs exciton sizes and considering if and how a more confined exciton can enhance the image charge effect, which has been reported by Tanaka et al. The calculation of Bohr radius and exciton binding energy provided in the revision shows that MA and Cs compounds have insignificant exciton size difference, unlikely to contribute to QCSEs of opposite signs. Furthermore, I find it intriguing that the authors observed a different field-dependence from what was observed by Tanaka and co-workers. A comprehensive and careful study on the interplay among image charge effect, cation polarization and electron and hole separation will be interesting and important to our understanding of exciton electronic structure in different hybrid perovskite materials.
2. The authors modified the fitting of the exciton absorption peaks using a 2D model. In addition, the authors were able to resolve the buried exciton peaks via defined ground-state bleach signals observed in transient absorption measurements. However, the pump pulse in TA experiments may induce an electric field that is significant enough to induce a transient EA signal. Therefore, the exciton resonances observed in TA may be a mixture of ground-state bleach and the exciton's EA signal, and thus rigorously, should not be referred to as ground-state bleach only.
3. The authors made a fair case as to why it is appropriate to use the Liptay equation to fit their exciton EA spectra.
4. The authors now provide refractive indices and dielectric functions obtained by modeling ellipsometry data. Checking if the interference effect changes the electroabsorption curves is critical and the authors data show that the reflectance geometry is likely to cause minor deviation from the true electroabsorption spectra. Furthermore, the authors demonstrate how the coupling of optical interference and electrostrictive effects affect the electroabsorption spectra. Indeed, lineshapes at higher than 2.8 eV and 2.5 eV induced by electrostriction and coupling of electrostriction and electroabsorption are absent from the authors' EA curves.

Reviewer #2 (Remarks to the Author):

I am happy to recommend publication of this revised version of the manuscript.

Reviewer #4 (Remarks to the Author):

Review of NCOMMS-18-04570A

There are three important claims in this paper

- 1) That the perovskite films show large Stark effects.
- 2) That they show a blue shift in some cases.
- 3) That the blue shift is a result of methylammonium (MA⁺) dipole re-orientation.

The first two claims are well supported. As the fourth reviewer I will not comment on them except for one thing. It is stated in the caption of figure 2 that the fits in the bottom row of figure 2, and by extension figure 3, are based on "on weighted sums of the zeroth-, first-, and second-derivatives of the excitonic absorption bands" . However, the actual weights used are not given anywhere I can find. Although some of the statements about the fits seem to be true "by eye" , it would be much better to put the weights in the figure. There seems to be plenty of room in the bottom row panels.

About the third claim, I don't think it's entirely proven, but I think the work done is sufficiently indicative that the paper can be published as is. The difference in signal from the Cs+ and MA+ containing films is quite strong evidence. I understand that additional DFT and Monte Carlo work has been requested and added by the authors. I have no problem with this work as carried out, but I do have a few comments about the presentation.

On the middle of page 11, it is noted that DFT indicates that a 5 to 10 degree rotation of the MA+ is enough to give the observed blue shift. Then at the bottom of page 11 it is said that the Monte Carlo simulations show an expected rotation of 1 degree at the field applied. It would appear from figure 4 b, that the DFT would predict only ~ 0.1 mV exciton binding energy shift for a 1 degree rotation.

Not enough to create a blue Stark shift. This mismatch is not commented on.

I personally think the difference is well within the uncertainty bands around the results from both techniques.

So the offset doesn't change the level of support given by the calculations.

However, I think some readers who take modeling results at face value may conclude that this offset weakens the argument that the MA+ dipole causes the blue stark shift.

Perhaps the authors can add a sentence about "order of magnitude" to avoid losing some readers at this point.

Also the sentence "The necessity of only a small perturbation is unsurprising given the likely activation energy of 10s of meV required for full rotation" is confusing. As I read it , it is circular reasoning. To me, the sentence says "assuming that the MA+ dipole is responsible for the blue shift, we would expect it to require a small rotational shift in the MA+ , because a large rotation is energetically expensive." To my mind, the point here is still to prove the MA+ rotation is responsible. Perhaps the authors

meant something like "That only a small net MA+ rotation is required to create the observed stark effect is promising , because a large rotation is energetically unlikely."?

Lastly, and at this late date I will not reject the paper on this basis, there seems to be very little actual data. There seem to be no repeat measurements on additional identical films. There don't seem to be control experiments to see if the same film, measured a second time, and a week later, gives the same signal. At least the bulk organic lead halide perovskites are notorious for showing different effects on the second measurement.

However, since the first 3 reviewers did not require more data, I think it would be unjust of me to require it now.

In conclusion, publish with the addition of the weights used for figure 2 and 3. The changes I have

mentioned regarding the MA+ dipole work are at the authors discretion.

Reviewer #1 (Remarks to the Author):

In this revised manuscript, the authors addressed the major issues in my last review with additional experiments, calculations and analyses listed below. Upon these revisions, I find their conclusion that the blue shift of the exciton resonance in the methylammonium (MA) containing perovskite is due to field induced reorientation of the MA cation well-supported. Therefore, I recommend this manuscript for publication in Nature Communications.

1. The authors addressed my concerns on attributing the exciton resonance blue-shift solely to the MA polarization without examining the MA and Cs exciton sizes and considering if and how a more confined exciton can enhance the image charge effect, which has been reported by Tanaka et al. The calculation of Bohr radius and exciton binding energy provided in the revision shows that MA and Cs compounds have insignificant exciton size difference, unlikely to contribute to QCSEs of opposite signs. Furthermore, I find it intriguing that the authors observed a different field-dependence from what was observed by Tanaka and co-workers. A comprehensive and careful study on the interplay among image charge effect, cation polarization and electron and hole separation will be interesting and important to our understanding of exciton electronic structure in different hybrid perovskite materials.

2. The authors modified the fitting of the exciton absorption peaks using a 2D model. In addition, the authors were able to resolve the buried exciton peaks via defined ground-state bleach signals observed in transient absorption measurements. However, the pump pulse in TA experiments may induce an electric field that is significant enough to induce a transient EA signal. Therefore, the exciton resonances observed in TA may be a mixture of ground-state bleach and the exciton's EA signal, and thus rigorously, should not be referred to as ground-state bleach only.

The exciton resonances observed in TA are no longer referred to as ground-state bleaches only.

3. The authors made a fair case as to why it is appropriate to use the Liptay equation to fit their exciton EA spectra.

4. The authors now provide refractive indices and dielectric functions obtained by modeling ellipsometry data. Checking if the interference effect changes the electroabsorption curves is critical and the authors data show that the reflectance geometry is likely to cause minor deviation from the true electroabsorption spectra. Furthermore, the authors demonstrate how the coupling of optical interference and electrostrictive effects affect the electroabsorption spectra. Indeed, lineshapes at higher than 2.8 eV and 2.5 eV induced by electrostriction and coupling of electrostriction and electroabsorption are absent from the authors' EA curves.

Reviewer #2 (Remarks to the Author):

I am happy to recommend publication of this revised version of the manuscript.

Reviewer #4 (Remarks to the Author):

Review of NCOMMS-18-04570A

There are three important claims in this paper

- 1) That the perovskite films show large Stark effects.
- 2) That they show a blue shift in some cases.
- 3) That the blue shift is a result of methylammonium (MA⁺) dipole re-orientation.

The first two claims are well supported. As the fourth reviewer I will not comment on them except for one thing. It is stated in the caption of figure 2 that the fits in the bottom row of figure 2, and by extension figure 3, are based on "on weighted sums of the zeroth-, first-, and second-derivatives of the excitonic absorption bands". However, the actual weights used are not given anywhere I can find. Although some of the statements about the fits seem to be true "by eye", it would be much better to put the weights in the figure. There seems to be plenty of room in the bottom row panels.

The fitting weights corresponding to the significant EA contributions have now been added to both of these figures.

About the third claim, I don't think it's entirely proven, but I think the work done is sufficiently indicative that the paper can be published as is. The difference in signal from the Cs⁺ and MA⁺ containing films is quite strong evidence. I understand that additional DFT and Monte Carlo work has been requested and added by the authors. I have no problem with this work as carried out, but I do have a few comments about the presentation.

On the middle of page 11, it is noted that DFT indicates that a 5 to 10 degree rotation of the MA⁺ is enough to give the observed blue shift. Then at the bottom of page 11 it is said that the Monte Carlo simulations show an expected rotation of 1 degree at the field applied. It would appear from figure 4 b, that the DFT would predict only ~0.1 mV exciton binding energy shift for a 1 degree rotation. Not enough to create a blue Stark shift. This mismatch is not commented on. I personally think the difference is well within the uncertainty bands around the results from both techniques. So the offset doesn't change the level of support given by the calculations. However, I think some readers who take modeling results at face value may conclude that this offset weakens the argument that the MA⁺ dipole causes the blue stark shift. Perhaps the authors can add a sentence about "order of magnitude" to avoid losing some readers at this point.

The following statement has been added to the main text discussion: "Along with the DFT calculations, they [Monte Carlo simulations] provide a clear order-of-magnitude estimate that only a small perturbation to the net alignment of the methylammonium dipoles is necessary and can be achieved with the fields in our experiments."

Also the sentence "The necessity of only a small perturbation is unsurprising given the likely activation energy of 10s of meV required for full rotation" is confusing. As I read it, it is circular reasoning. To me, the sentence says "assuming that the MA⁺ dipole is responsible for the blue shift, we would expect it to require a small rotational shift in the MA⁺, because a large rotation

is energetically expensive." To my mind, the point here is still to prove the MA+ rotation is responsible. Perhaps the authors meant something like "That only a small net MA+ rotation is required to create the observed stark effect is promising, because a large rotation is energetically unlikely."?

We have amended this statement to improve its clarity as follows: "The necessity of only a small perturbation is promising given that a full alignment would be energetically unlikely^{28,30,39,51}".

Lastly, and at this late date I will not reject the paper on this basis, there seems to be very little actual data. There seem to be no repeat measurements on additional identical films. There don't seem to be control experiments to see if the same film, measured a second time, and a week later, gives the same signal. At least the bulk organic lead halide perovskites are notorious for showing different effects on the second measurement. However, since the first 3 reviewers did not require more data, I think it would be unjust of me to require it now.

We now include the following supplementary figure showing electro-absorption spectra measured for an $n = 3$ methylammonium device following fabrication and eight months after.

Minor differences in the intensity of the EA signals is observable, which we attribute to material degradation.

In conclusion, publish with the addition of the weights used for figure 2 and 3. The changes I have mentioned regarding the MA+ dipole work are at the authors discretion.